https://doi.org/10.1038/s41467-021-26867-8　　**OPEN**

# Discovery of putative tumor suppressors from CRISPR screens reveals rewired lipid metabolism in acute myeloid leukemia cells

W. Frank Lenoir [1,2], Micaela Morgado[2], Peter C. DeWeirdt[3], Megan McLaughlin [1,2], Audrey L. Griffith [3], Annabel K. Sangree [3], Marissa N. Feeley[3], Nazanin Esmaeili Anvar[1,2], Eiru Kim[2], Lori L. Bertolet[2], Medina Colic [1,2], Merve Dede[1,2], John G. Doench [3] & Traver Hart [2,4✉]

CRISPR knockout fitness screens in cancer cell lines reveal many genes whose loss of function causes cell death or loss of fitness or, more rarely, the opposite phenotype of faster proliferation. Here we demonstrate a systematic approach to identify these proliferation suppressors, which are highly enriched for tumor suppressor genes, and define a network of 145 such genes in 22 modules. One module contains several elements of the glycerolipid biosynthesis pathway and operates exclusively in a subset of acute myeloid leukemia cell lines. The proliferation suppressor activity of genes involved in the synthesis of saturated fatty acids, coupled with a more severe loss of fitness phenotype for genes in the desaturation pathway, suggests that these cells operate at the limit of their carrying capacity for saturated fatty acids, which we confirm biochemically. Overexpression of this module is associated with a survival advantage in juvenile leukemias, suggesting a clinically relevant subtype.

[1] The University of Texas MD Anderson Cancer Center UTHealth Graduate School of Biomedical Sciences; The University of Texas MD Anderson Cancer Center, Houston, TX, USA. [2] Department of Bioinformatics and Computational Biology, The University of Texas MD Anderson Cancer Center, Houston, TX, USA. [3] Genetic Perturbation Platform, Broad Institute of MIT and Harvard, Cambridge, MA, USA. [4] Department of Cancer Biology, The University of Texas MD Anderson Cancer Center, Houston, TX, USA. ✉email: traver@hart-lab.org

Gene knockouts are a fundamental tool for geneticists, and the discovery of CRISPR-based genome editing[1] and its adaptation to gene knockout screens has revolutionized mammalian functional genomics and cancer targeting[2–8]. Hundreds of CRISPR/Cas9 knockout screens in cancer cell lines have revealed background-specific genetic vulnerabilities[9–13], providing guidance for tumor-specific therapies and the development of targeted agents. Although lineage and mutation state are powerful predictors of context-dependent gene essentiality, variation in cell growth medium and environment can also drive differences in cell state, particularly among metabolic genes[14,15], and targeted screening can reveal the genetic determinants of metabolic pathway buffering[16,17].

The presence and composition of metabolic and other functional modules in the cell can also be inferred by integrative analysis of large numbers of screens. Correlated gene knockout fitness profiles, measured across hundreds of screens, have been used to infer gene function and the modular architecture of the human cell[18–21]. Data-driven analysis of correlation networks reveals clusters of functionally related genes whose emergent essentiality in specific cell backgrounds is often unexplained by the underlying lineage or mutational landscape[21]. Interestingly, in a recent study of paralogs whose functional buffering renders them systematically invisible to monogenic CRISPR knockout screens[22,23], it was shown that the majority of context-dependent essential genes are constitutively expressed in cell lines[23]. Collectively these observations suggest that there is much unexplained variation in the genetic architecture, and emergent vulnerability, of tumor cells.

Building human functional interaction networks from correlated gene knockout fitness profiles in cancer cells is analogous to generating functional interaction networks from correlated genetic interaction profiles in S. cerevisiae[24–27]. The fundamental difference between the two approaches is that, in yeast, a massive screening of pairwise gene knockouts in a single yeast strain was conducted in order to measure genetic interaction—a dual-knockout phenotype more or less severe than that expected by the combination of the two genes independently. In coessentiality networks, CRISPR-mediated single-gene knockouts are conducted across a panel of cell lines that sample the diversity of cancer genotypes and lineages. Digenic perturbations in human cells, a more faithful replication of the yeast approach, are possible with Cas9 and its variants, but library construction, sequencing, and positional biases can be problematic[16,28–34]. Recently, we showed that an engineered variant of the Cas12a endonuclease, enCas12a[35], could efficiently perform multiplex gene knockouts[34], and we demonstrated its effectiveness in assaying synthetic lethality between targeted paralogs[23]. These developments in principle enable researchers to measure how biological networks vary across backgrounds, a powerful approach for deciphering complex biology[24,36,37].

CRISPR perturbations in human cells can result in loss-of-function alleles that increase as well as a decrease in vitro proliferation rates; faster proliferation is an extreme rarity in yeast knockouts. These fast-growers can complicate predictions of genetic interaction[29] and confound pooled chemoresistance screens[38]. However, there is no broadly accepted method of identifying these genes from CRISPR screens.

In this work, we describe the development of a method to systematically classify genes whose knockout provides a proliferation advantage in vitro. We observe that genes that confer proliferation advantage are typically tumor suppressor genes and that they show the same modularity and functional coherence as context-dependent essential genes. Moreover, we discover a module that includes several components of the glycerolipid biosynthesis pathway that slows cell proliferation in a subset of acute myeloid leukemia (AML) cell lines. We show a rewired genetic interaction network using enCas12a multiplex screening, and find strong genetic interactions corroborated by clinical survival data. A putative tumor-suppressive role for glycerolipid biosynthesis is noteworthy considering this process is thought to be required to generate biomass for tumor cell growth, and inhibitors targeting this pathway are currently in clinical trials[39,40].

## Results

**Identifying proliferation-suppressor signatures.** We previously observed genes whose knockout leads to overrepresentation in pooled library knockout screens. These genes, which we term proliferation-suppressor genes (PSG), exhibit positive selection in fitness screens, a phenotype opposite that of essential genes. As expected, many PSG are known tumor suppressor genes; for example, TP53 and related pathway genes CDKN1A, CHEK2, and TP53BP1 show positive selection in select cell lines (Fig. 1a). Detection of these genes as outliers is robust to the choice of CRISPR analytical method, as we tested BAGEL2[41,42], CERES[10], JACKS[43], and mean log-fold change (LFC) of gRNA targeting each gene (Supplementary Fig. 1a–d). Unlike core-essential genes, PSG are highly context-specific: TP53 knockout shows positive LFC only in cell lines with wild-type TP53 (Fig. 1b), and PTEN knockout shows the PS phenotype only in $PTEN^{wt}$ backgrounds (Fig. 1c). These observations are consistent with the knockout phenotypes of known tumor suppressor genes (TSG) in cell lines: in wild-type cells, TSG knockout increases the proliferation rate in cell culture, but when cell lines are derived from tumors where the TSG is already lost or non-functional, gene knockout has no effect. TSG are therefore context-specific PSG, but it is not necessarily the case that genes with a proliferation-suppressor phenotype in vitro act as TSG in vivo; proliferation suppressors are at best putative tumor suppressors in the absence of confirmatory data from tumor profiling.

Though detection of PSG is possible using existing informatics pipelines, several factors complicate a robust detection of these genes. There is no accepted threshold for any algorithm we considered to detect PSG, since all were optimized to classify essential genes. A related second issue is that cell line screens show a wide range of variance in LFC distributions, making robust outlier detection challenging (Supplementary Fig. 1e, f). Third, the signatures are strongly background-dependent, as demonstrated by PTEN and TP53. Finally, there is no consistent expectation for whether or how many putative tumor suppressor genes are present in a given cell line.

To address this gap, we developed a method to account for variability in fold-change distributions between screens. Our approach uses a Gaussian mixture model (K = 2) to estimate each screen's distribution of gene-level LFC scores (Fig. 1a). Mixed distribution models have previously been used to identify distinctions between populations of essential and nonessential fitness genes in CRISPR screens[44]. For the K = 2 mixture model, the more negative distribution (Fig. 1a, red) is generally essential genes, while the higher, narrower peak around zero (Fig. 1a, blue), models the large population of knockouts with no fitness phenotype. We used this second model to calculate a Z-score (hereafter referred to as the "mixed Z-score") for all gene-level mean fold changes in each cell line. This approach normalizes variance (Supplementary Fig. 1e, f) across LFC distributions in different cell lines, with negative Z-scores indicating essential genes and positive Z-scores representing PSG phenotypes.

To evaluate the effectiveness of this mixed Z-score approach, we used COSMIC[45,46] tumor suppressor genes as a true positive reference set, and we combined COSMIC-defined oncogenes (removing dual-annotated tumor suppressors) with our

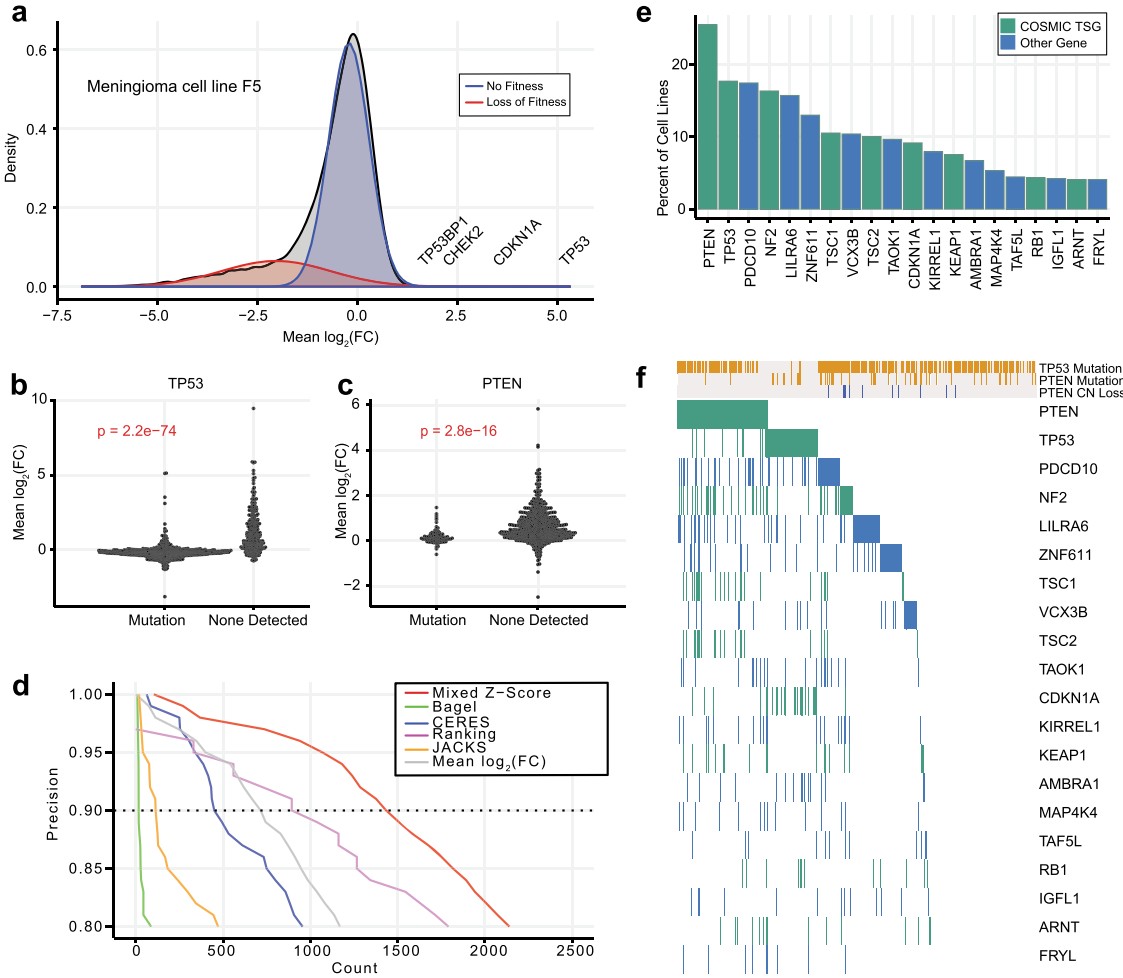

**Fig. 1 Discovery of proliferation-suppressor genes. a** Fold-change distribution of a typical CRISPR knockout screen has a long left tail of essential genes, and a small number of genes whose knockout increases fitness (proliferation-suppressor genes, "PSG"). A two-component Gaussian mixture model (red, blue) models this distribution. **b**, **c** Fold change of common tumor suppressors across 808 cell lines (P values, two-sided Wilcoxon rank-sum test). **d** Precision vs. recall of mixed Z-score and other CRISPR analysis methods. Dashed line, 90% precision (10% FDR). **e** Fraction of cell lines in which known tumor suppressors (green) or other genes (blue, not defined as TSG by COSMIC) are classified as PS genes at 10% FDR. **f** Presence of each known TSG across 808 cell lines, vs. cell genetic background. Gold, mutation present; gray, absent. Green or blue, following color scheme in (**e**), gene is classified as a proliferation suppressor.

previously-specified set of nonessential genes as a true negative reference set[7,47]. Since there is no expectation for the presence of a consistent set of PSG across cell lines, we analyzed each of the 808 cell lines from the Avana 2020Q4 data release independently[10,48,49] calculating gene-level scores on each cell line individually and then combining all scores into a master list of $808 \times 18k = 14.6$ million gene-cell line observations (Supplementary Data 1). Moreover, since there is also no expectation that all COSMIC TSG would be detected cumulatively across all cell lines, we judged that traditional recall metrics (e.g., percentage of the reference set recovered) were inappropriate. We, therefore, defined recall as the total number of TSG-cell line observations. Using this evaluation scheme, the mixed Z-score approach outperforms comparable methods by a substantial margin, classifying more than 722 PS-cell line instances at a 10% false discovery rate (FDR) (Fig. 1d). This is ~50% more putative PSG than the closest alternative, a nonparametric rank-based approach, at the same FDR. BAGEL[41,42], a supervised classifier of essential genes, performed worst at TSG, and the raw mean LFC approach also fared poorly, highlighting the need for variance normalization across experiments. We applied this 10% FDR threshold for all subsequent analyses.

Common tumor suppressor genes *PTEN* and *TP53* were observed in ~25% and ~18% of cell lines, respectively (Fig. 1e), with other well-known TSG appearing less frequently. Among 309 COSMIC TSGs for which we have fitness profiles (representing 1.7% of all 18 k genes), we find that 116 (37.5%) of these genes occur as proliferation suppressors at least once (Supplementary Data 2) and make up 24.4% of total proliferation-suppressor calls (Supplementary Fig. 2a, b), a 14-fold enrichment. All of the known TSG hits come from just 504 of the 808 cell lines (62.4%) in which proliferation-suppressor hit calls were identified (Fig. 1f), yet we did not observe a bias toward particular tissues: in every lineage, most cell lines carried at least one PSG (Supplementary Fig. 1g).

To further validate our approach, we compared the set of TSGs in our PSG hits to other molecular profiling data. When identified as a proliferation suppressor, 53% of the 116 TSGs demonstrate higher mean mRNA expression relative to backgrounds where the same TSG is not a proliferation suppressor (Supplementary Data 2). Similarly, 96.6% of the 116 TSGs, when classified as a proliferation suppressor, demonstrate a lower frequency of nonsilent mutations compared to backgrounds where the TSG is not a hit (Supplementary Data 2). These observations were not

restricted to COSMIC TSGs however, as this was the case for all PSG hit calls of genes against non-PSG hit calls (Supplementary Fig. 2c, d). Copy-number comparisons did not suggest major distinctions between PSG vs. non-PSG calls (Supplementary Fig. 2e), however, there did appear to be more variation in PSG observations, possible stemming from smaller grouped sample sizes. Together, these observations confirm the reliability of our method to detect genes whose knockout results in faster cell proliferation, and that, analogous to essential genes, these genes must be expressed and must not harbor a loss-of-function mutation in order to elicit this phenotype.

We attempted to corroborate our findings using a second CRISPR dataset of 342 cell line screens from Behan et al.[13], including >150 screens in the same cell lines as in the Avana data. However, these screens were conducted over a shorter timeframe than the Avana screens (14 vs. 21 days), giving less time for both positive and negative selection signals to appear (see "Methods" for a detailed discussion). As a result, when we compared cell lines screened by both groups, the Avana data yielded many more TSG hits (Supplementary Fig. 3a). While most of these do not meet our threshold for PSG in the Sanger data, hits at our 10% FDR threshold across all Avana screens are strongly biased toward positive mixed $Z$-scores in the Sanger screens (Supplementary Fig. 3b), consistent with a weaker signal of positive selection as a result of the shorter assays rather than a lack of robustness in the screens[49].

**Discovering pathways modulating cell growth with a proliferation-suppressor co-occurrence network**. Although known TSG act as PSG in only a subset of cell lines, we observed patterns of co-occurrence among functionally related genes. *PTEN* co-occurs with mTOR regulators *NF2*[50] ($P < 3 \times 10^{-11}$, Fisher's exact test) and the *TSC1/TSC2* complex ($P$ values both $<7 \times 10^{-13}$)[51], as well as Programmed Cell Death 10 (*PDCD10*)[52], a proposed tumor suppressor[7,53] (Fig. 2a). The p53 regulatory cluster (*TP53*, *CDKN1A*, *CHECK2*, *TP53BP1*) also exhibited a strong co-occurrence pattern that was independent of the mTOR regulatory cluster (Fig. 2a). mTOR[54] and cell cycle checkpoint genes[55,56] have been heavily linked to cancer development, given their roles in controlling cell growth and proliferation, and thus have been the focus of studies characterizing patient genomic profiles to identify common pathway alterations[57,58].

The modularity of mTOR regulators and TP53 regulators demonstrates pathway-level proliferation-suppressor activity. This reflects the concept of genes with correlated fitness profiles indicating the genes operate in the same biochemical pathway or biological process[19,21,59,60]. However, the sparseness of PSG, coupled with their smaller effect sizes, renders correlation networks relatively poor at identifying modules of genes with proliferation-suppressor activity. In order to identify such modules, we developed a PSG network (Supplementary Data 3) based on the statistical overrepresentation of co-occurring PSG (Fig. 2b); see "Methods" for details. This approach yields a network of 145 genes containing 462 edges in disconnected clusters; only 8 clusters have 3 or more genes (Fig. 2c and Supplementary Fig. 4c). Of these 462 edges, 74 (16.0%, empirical $P < 10^{-4}$) are present in the HumanNet[61] functional interaction network (Supplementary Fig. 4a, b), ~eightfold more than expected from random sampling, indicating high functional coherence between connected genes. The network recovers the PTEN and TP53 modules as well as the Hippo pathway, the aryl hydrocarbon receptor complex (AHR/ARNT), the mTOR-repressing GATOR1 complex, the STAGA chromatin remodeling complex, JAK-STAT signaling, and the gamma-secretase complex (Fig. 2c and Supplementary 4c), all of which have been associated

with tumor suppressor activity. The functional coherence and biological relevance of the PSG co-occurrence network further validates the approach taken and establishes this dataset as a resource for exploring putative tumor suppressor activity in cell lines and tumors.

**Variation in fatty acid metabolism in AML cells**. In addition to the known tumor suppressors, we observed a large module containing elements of several fatty acid (FA) and lipid biosynthesis pathways (Fig. 2c). Interestingly, while there does not appear to be a strong tissue specificity signature for most clusters (Fig. 2c), the fatty acid metabolism cluster shows a strong enrichment for AML cell lines ($P = 1.5 \times 10^{-4}$). AML, like most cancers, typically relies on increased glucose consumption for energy and diversion of glycolytic intermediates for the generation of biomass required for cell proliferation. Membrane biomass is generated by phospholipid biosynthesis that uses fatty acids as building blocks, with FA pools replenished by some combination of triglyceride catabolism, transporter-mediated uptake, and de novo synthesis via the *ACLY/ACACA/FASN* palmitate production pathway using citrate precursor diverted from the TCA cycle. Indeed, the role of lipid metabolism in AML progression is indicated by changes in serum lipid content[62], in particular for long-chain saturated fatty acids that are the terminal product of the FAS pipeline. Inhibition of FA synthesis is therefore an appealing chemotherapeutic intervention[63,64] and FASN inhibitors are currently undergoing clinical trials for the treatment of solid tumors and metabolic diseases[40]. The observation that knocking out FAS pathway genes results in faster proliferation in some AML cells, and their signature as putative tumor suppressor genes, is therefore very unexpected.

To learn whether additional elements of lipid metabolism were associated with the FAS cluster, we examined the differential correlation of mixed $Z$-scores in AML cells. We and others have shown that genes with correlated gene knockout fitness profiles in CRISPR screens are likely to be involved in the same biological pathway or process ("co-functional")[18–21], analogous to correlated genetic interaction profiles in yeast[25,26,65]. Strikingly, all gene pairs within the fully connected clique in the FAS cluster (containing genes *FASN*, *ACACA*, *GPAT4*, *CHP1*, *GPI CERS6*, *PCGF1*, Fig. 2c) had a median Pearson correlation coefficient (PCC) of 0.76 in the 23 AML cell lines (range 0.63–0.95, Fig. 3a, red), compared to the median correlation of 0.05 in the remaining 785 cell lines (range −0.11–0.62, with the highest correlation between *FASN* and *ACACA*, adjacent enzymes in the linear palmitate synthesis pathway; Fig. 3a, gray). These high differential Pearson correlation coefficients (dPCC) suggest that variation in lipid metabolism is pronounced in AML cells[66].

We sought to explore whether this difference in correlation identified other genes that might give insight into metabolic rewiring in AML. We first removed noisy data by filtering for high-quality screens (Cohen's $D > 2.5$, recall >60%[42]), leaving 659 cell lines, including 17 AML cell lines. Calculating a global difference between PCC of all gene pairs in all 17 AML and in the remaining 642 cell lines yielded many gene pairs whose dPCC appeared indistinguishable from random sampling (Supplementary Fig. 5a, b). To filter these, we calculated empirical $P$ values for each gene pair. We randomly selected 17 cell lines from the pool of all screens, calculated PCC for all gene pairs in the selected and remaining lines, and calculated dPCC from these PCC values (Fig. 3b). We repeated this process 1000 times to generate a null distribution of dPCC values for each gene pair, against which a $P$ value could be computed (Fig. 3c, d).

Expanding the set to a filtered list of genes whose correlation with a gene in the FAS clique showed significant change in AML

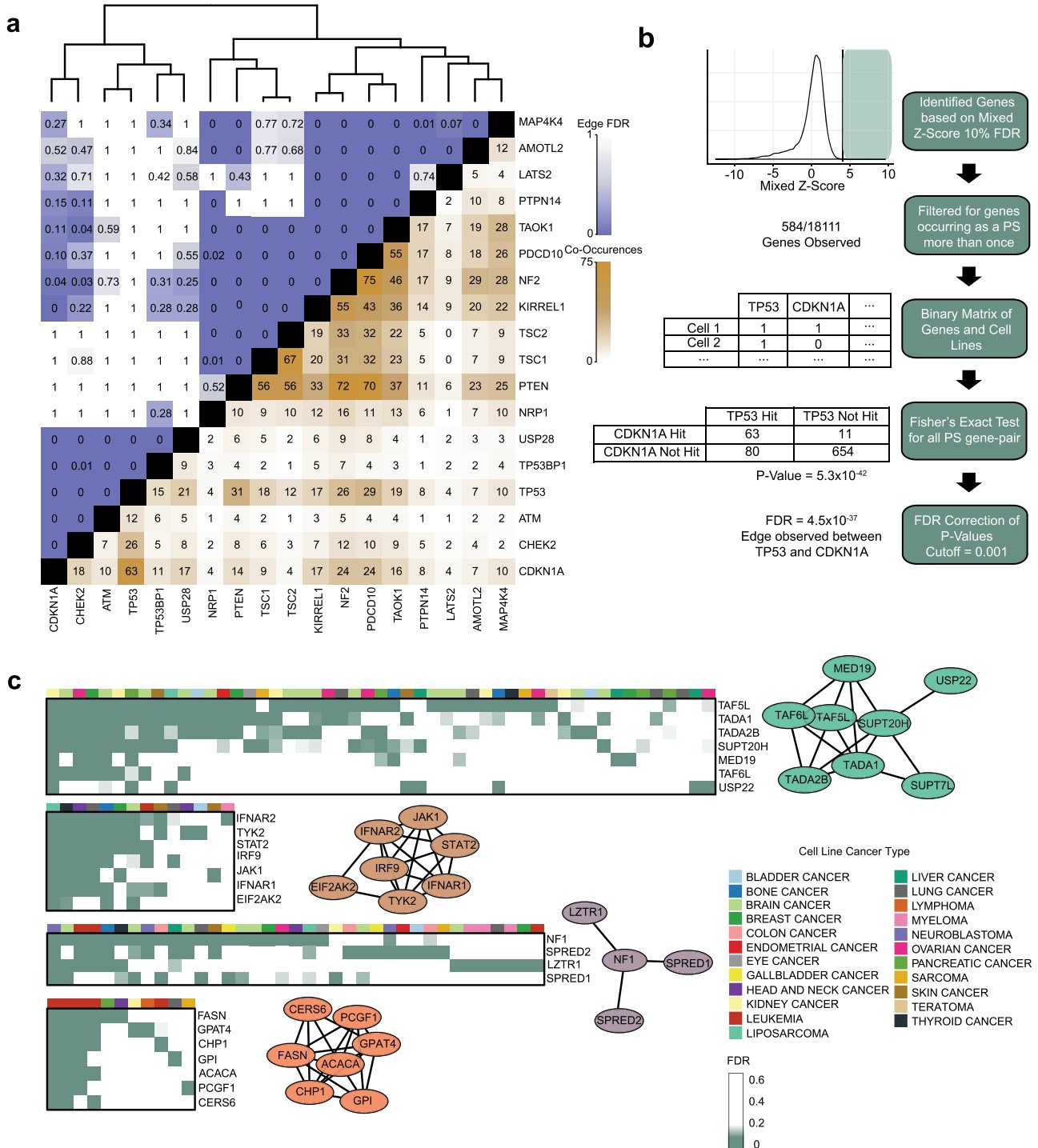

**Fig. 2 Co-occurrence of PSG. a** Co-occurrence/mutual exclusivity of common TSG as PSG in CRISPR screens. Brown, number of cell lines in which two genes co-occur as PSG at 10% FDR. Blue, FDR of co-occurrence. Hierarchical clustering delineates functional modules. **b** Pipeline for building the co-PS network. **c** Examples from the Co-PS network. Nodes are connected by edges at FDR < 0.1%. Heatmaps indicate the presence of PSG vs. cell lineage.

cells ($P < 0.001$; see "Methods") yielded a total of 106 genes, including the 7 genes in the clique (Fig. 3e) plus holocarboxylase synthetase (*HLCS*), which biotinylates and activates acetyl-CoA-carboxylase, the protein product of *ACACA*, as well as glycolysis pathway genes *PGP* and *HK2*. Interestingly, about half of the genes showed significantly increased anticorrelation with the FAS cluster, indicating genes preferentially essential where the FAS genes act as proliferation suppressors (Fig. 3e). These genes include fatty acid desaturase (*SCD*), which operates directly

downstream from *FASN/ACACA* to generate monounsaturated fatty acid species, and sterol-regulatory element-binding transcription factor 1 (*SREBF1*), the master regulatory factor for lipid homeostasis in cells.

Clustering the AML cells lines according to these high-dPCC genes reveals two distinct subsets of cells. The FAS cluster and its correlates show a strong proliferation-suppressor phenotype in four cell lines, NB4, MV411, MOLM13, and THP1. The remaining thirteen AML cell lines show negligible to weakly

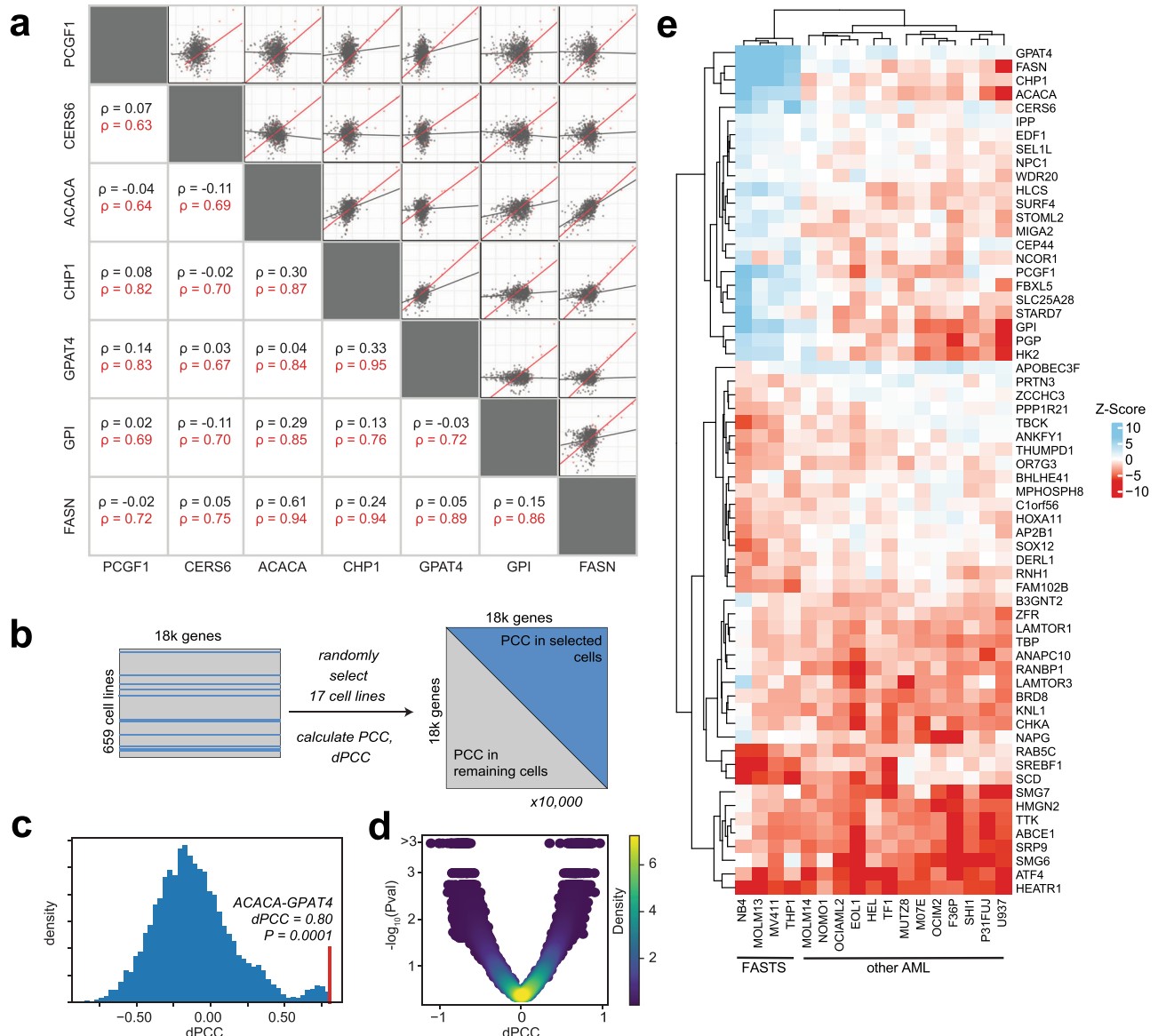

**Fig. 3 Differential network analysis of fatty acid synthesis module. a** Among genes in the FAS module, Pearson correlation coefficients of mixed $Z$-score profiles are substantially higher in AML cells (red) than in other cells (gray). **b** Significance testing of differential PCC (dPCC) involves quality filtering of Avana data ($n = 659$ cell lines, including 17 AML cell lines), building a null distribution by randomly selecting 17 cell lines, and calculating PCC between all gene pairs in the selected cells and the remaining cells. **c** After 10,000 repeats, a null distribution is generated for each pair, and a $P$ value (permutation test, $n = 10,000$) is calculated for the observed AML vs. other dPCC. **d** Volcano plot of dPCC vs. $P$ value for all genes in the Co-PS cluster. **e** Heatmap of mixed $Z$-score for 17 AML cell lines in selected genes with high |mixed Z| and high |dPCC|. Clustering of cell lines indicates the putative fatty acid synthesis/tumor suppressor (FASTS) subtype.

essential phenotypes when these genes are knocked out. The anticorrelated genes, including *SCD* and *SREBF1*, show heightened essentiality in these same cell lines. Together these observed shifts in gene knockout fitness indicate that this subset of AML cells has a substantial metabolic rewiring. Because these cells share a genetic signature among fatty acid synthesis pathway genes that is consistent with tumor suppressors, we call these cell lines F̲atty A̲cid S̲ynthesis/T̲umor S̲uppressor (FASTS) cells (Fig. 3e).

**Cas12a-mediated genetic interaction screens confirm rewired lipid metabolism.** We sought to confirm whether gene knockout confers improved cell fitness, and to gather some insight into why some AML cells show the FASTS phenotype and others do not.

Genetic interactions have provided a powerful platform for understanding cellular rewiring in model organisms, and we sought to apply this approach to deciphering the FASTS phenotype. We designed a CRISPR screen that measures the genetic interactions between eight selected "query genes" and ~100 other genes ("array genes"). The query genes include *FASN* and *ACACA*, from the cluster of proliferation-suppressor genes, as well as lipid homeostasis transcription factor *SREBF1*, anticorrelated with the FAS cluster in the differential network analysis, and uncharacterized gene *c12orf49*, previously implicated in lipid metabolism by coessentiality[21] and a recent genetic interaction study[60]. Additional query genes include control tumor suppressor genes *TP53* and *PTEN* and control context-dependent essential genes *GPX4* and *PSTK* (Fig. 4a). The array of genes include two to three genes each from several metabolic pathways, including various branches of

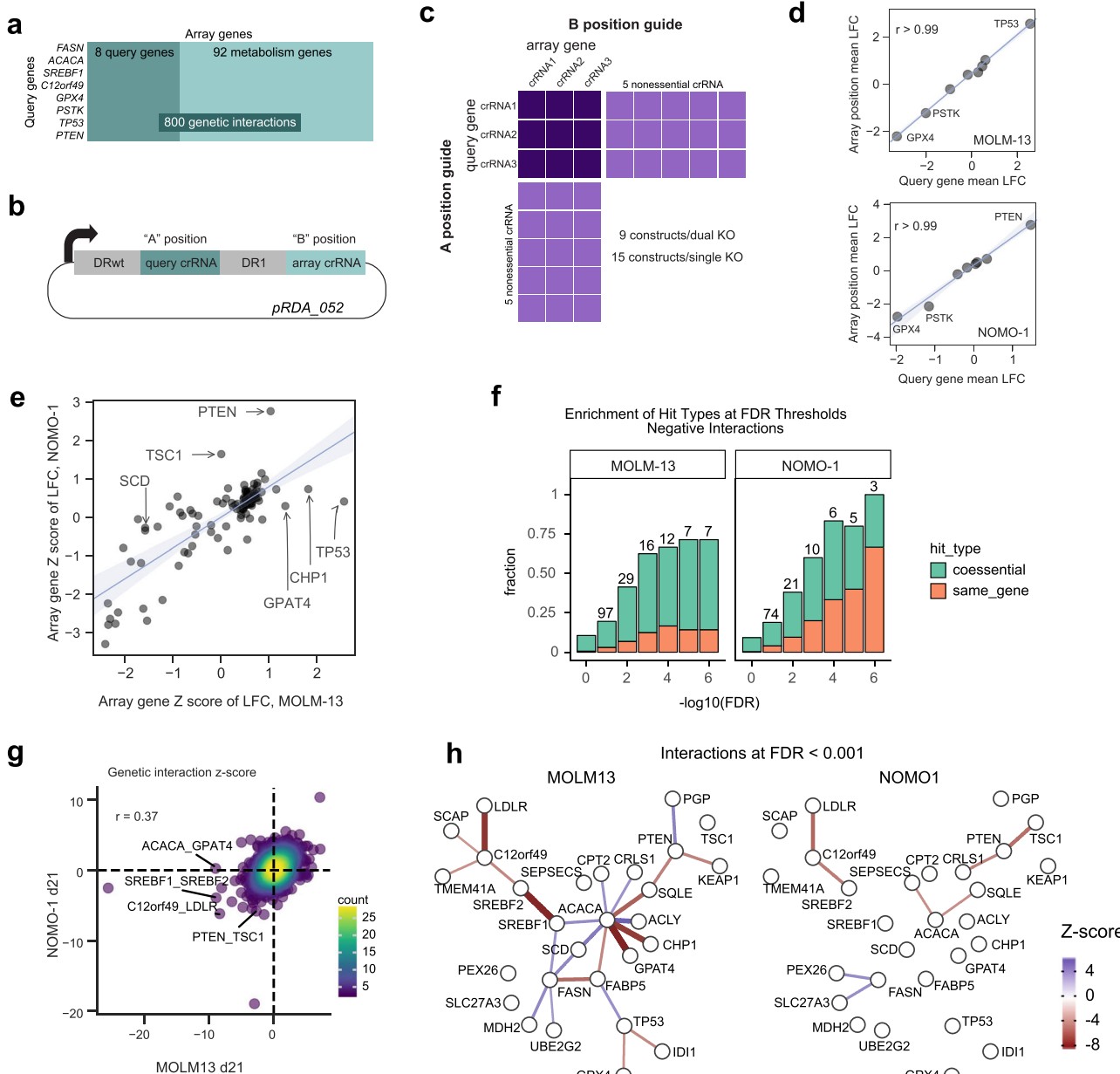

**Fig. 4 Genetic interactions reveal a rewired lipid biosynthesis pathway in FASTS cells. a** Genetic interaction screen targets eight query genes, selected from FASTS cluster and dPCC analysis, and 100 array genes sampling lipid metabolism pathways, for a total of 800 pairwise knockouts. **b** Library design uses a dual-guide enCa12a expression vector which targets the query gene in the "A" position and array gene in the "B" position. **c** Overall library design includes three crRNA/gene plus control crRNA targeting nonessential genes. Single-knockout constructs (target gene paired with nonessential controls) allow accurate measurement of single-knockout fitness. **d** Considering single-knockout fitness of query genes in the "A" and "B" position of the crRNA expression vector shows no position effects in the two cell lines screened (MOLM13, NOMO1). LFC log-fold change. **e** Single-knockout fitness (Z-score of mean LFC) is highly consistent between MOLM13 and NOMO1, but reveals background-specific PS genes. **f** Enrichment among GI for coessential and self-interacting genes. Self-interactions among genes that show single-knockout fitness phenotypes are expected, reflecting the quality of GI observations. **g** Global comparison of MOLM13, NOMO1 genetic interaction Z-scores. **h** Network view of interactions in each background shows rewiring in MOLM13 FASTS cells.

lipid biosynthesis, glycolysis and glutaminolysis, oxphos, peroxisomal and mitochondrial fatty acid oxidation. We include the query genes in the array gene set (Fig. 4a) to test for screen artifacts and further add control essential and nonessential genes to measure overall screen efficacy (Supplementary Data 4 and 5).

We used the enCas12a CRISPR endonuclease system to carry out multiplex gene knockouts[35]. We used a dual-guide enCas12a design, as described in DeWeirdt et al.[34], that allows for the construction of specific guide pairs through pooled oligonucleotide

synthesis (Fig. 4b). The library robustly measures single-knockout fitness by pairing three Cas12a crRNA per target gene each with five crRNA targeting nonessential genes[7,47] ($n = 15$ constructs for single-knockout fitness), and efficiently assays double-knockout fitness by measuring all guides targeting query-array gene pairs ($n = 9$) (Fig. 4c and Supplementary Data 5). Using this efficient design and the endogenous multiplexing capability of enCas12a, we were able to synthesize a library targeting 800 gene pairs with a single 12 k oligonucleotide array.

We screened one AML cell line from the FASTS subset, MOLM13, and a second one with no FAS phenotype, NOMO1, collecting samples at 14 and 21 days after transduction with a five-day puromycin selection (Supplementary Data 6 and 7). Importantly, by comparing the mean log-fold change of query gene knockouts in the "A" position vs. the same genes in the "B" position of the dual-knockout vector, we find no positional bias in the multiplex knockout constructs (Fig. 4d), consistent with our previous findings[23,34]. Single-knockout fitness measurements effectively segregated known essential genes from nonessentials, confirming the efficacy of the primary screens (Supplementary Fig. 6). Context-dependent fitness profiles are consistent with the cell genotypes, with PTEN and TSC1 showing positive selection in PTEN[wt] NOMO1 cells and TP53 being a strong PS gene in P53[wt] MOLM13 cells. Strikingly, CHP1 and GPAT4 are the next two top hits in MOLM13, confirming their proliferation-suppressor phenotype (Fig. 4e), while neither shows a phenotype in NOMO1. Together these observations validate the enCas12a-mediated multiplex perturbation platform, confirm the ability of CRISPR knockout screens to detect proliferation suppressors and corroborate the background-specific fitness-enhancing effects of genes from the FAS cluster.

To measure genetic interactions, we fit a linear regression for each guide between the combination LFCs and the single-guide LFCs, Z-scoring the residuals from this line, and combining across all guides targeting the same gene pair (Supplementary Fig. 6 and Supplementary Data 7). Here, positive genetic interaction Z-scores reflect greater fitness than expected and negative Z-scores represent lower than expected based on the single-gene knockouts independently, similar to the methodology applied in a recent survey of genetic interactions in cancer cells using multiplex CRISPR perturbation[33] (see "Methods"). Gene self-interactions (when the same gene is in the A and B position, Fig. 4d) should therefore be negative for proliferation suppressors and positive for essentials (Fig. 4f, g and Supplementary Fig. 6). Overall, genetic interaction Z-scores in the two cell lines showed moderate correlation (Fig. 4g), and previously reported synthetic interactions between C12orf49 and low-density lipoprotein receptor LDLR[17] and between SREBF1 and its paralog SREBF2[17] are identified in both cell lines (Supplementary Fig. 6f, g).

In contrast with the interactions found in both cell lines, background-specific genetic interactions reflect the genotypic and phenotypic differences between the cells. The negative interaction between tumor suppressor PTEN and mTOR repressor TSC1 in PTEN[wt] NOMO1 cells is consistent with their epistatic roles in the mTOR regulatory pathway. Both genes show positive knockout fitness in NOMO1 (Fig. 4e) but their dual knockout does not provide an additive growth effect, resulting in a suppressor interaction with a negative Z-score (Fig. 4g, h). Similarly, suppressor genetic interactions between ACACA and downstream proliferation-suppressor genes CHP1 and GPAT4 are pronounced in MOLM13 cells, consistent with epistatic relationships in a linear biochemical pathway (Fig. 4h). These interactions are not replicated with query gene FASN, but both FASN and ACACA show negative interactions with fatty acid transport gene FABP5 and positive interactions with SREBF1 and SCD, the primary desaturase of long-chain saturated fatty acids. All of these interactions are absent in NOMO1, demonstrating the rewiring of the lipid biosynthesis genetic interaction network between these two cell types (Fig. 4h).

**FASTS signature predicts sensitivity to saturated fatty acids.** The significant differences in the single- and double-knockout fitness signatures between the two cell lines suggest a major rewiring of lipid metabolism in these cells. CHP1 and GPAT4 are reciprocal top correlates in the Avana coessentiality network ($r = 0.43$, $P = 2.5 \times 10^{-34}$), strongly predicting gene co-functionality[21]. Two recent studies characterized the role of lysophosphatidic acid acyltransferase GPAT4 in adding saturated acyl moieties to glycerol 3-phosphate, generating lysophosphatidic acid (LPA) and phosphatidic acid (PA), the precursors for cellular phospholipids and triglycerides, and further discovered CHP1 as a key regulatory factor for GPAT4 activity[67,68]. Within hematological cancer cell lines, the coessentiality network is significantly restructured, with the ACACA/FASN module correlated with SCD in most backgrounds ($r = 0.35$, $P < 10^{-18}$) but strongly anticorrelated in 36 blood cancer cell lines ($r = -0.52$, $P < 10^{-3}$, Fig. 3e). The magnitude of this change in correlation is ranked #8 out of 31 million gene pairs (see "Methods"). In contrast, ACACA and FASN are weakly correlated with CHP1 in most tissues but strongly correlated in AML, with underlying covariation largely driven by the PS phenotype in FASTS cells (Fig. 3e). This pathway sign reversal is confirmed in the single-knockout fitness observed in our screens: SCD is strongly essential in MOLM13 but not in NOMO1 (Fig. 4e).

Collectively these observations make a strong prediction about the metabolic processing of specific lipid species. Faster proliferation upon knockout of genes related to saturated fatty acid processing, coupled with increased dependency on fatty acid desaturase gene SCD (Fig. 5a), suggests that these cells are at or near their carrying capacity for saturated fatty acids. To test this prediction, we exposed three FASTS cell lines and four other AML cell lines to various species of saturated and unsaturated fatty acids. FASTS cells showed significantly increased apoptosis in the presence of 200 μM palmitate (Fig. 5b, c) while no other species of saturated or unsaturated fatty acid showed similar differential sensitivity. In addition, analysis of metabolic profiles of cells in the Cancer Cell Line Encyclopedia[69,70] showed that saturated acyl chains are markedly overrepresented in triacylglycerol (TAG) in FASTS cells (Fig. 5d), in contrast with other lipid species measured (Supplementary Fig. 7). Palmitate-induced lipotoxicity has been studied in many contexts—and importantly, the role of GPAT4 and CHP1 in mediating lipotoxicity was well described recently[67,68]—but to our knowledge, this is the only instance of a genetic signature that predicts liposensitivity.

**Prognostic signature for FASTS genes.** To explore whether the FASTS phenotype has clinical relevance, we compared our results with patient survival information from public databases. Using genetic characterization data from CCLE[69], we did not find any lesion which segregated FASTS cells from other CD33 + AML cells (Fig. 6a), so no mutation is nominated to drive a FASTS phenotype in vivo. Instead, we explored whether variation in gene expression was associated with patient outcomes. We included genes in the core FASTS module as well as genes with strong genetic interactions with ACACA/FASN in our screen (Fig. 6a). To select an appropriate cohort for genomic analysis, we first considered patient age. Although AML presents across every decade of life, patients from whom FASTS cell lines were derived are all under 30 years of age (sources of other AML cells ranged from 6 to 68 years; Fig. 6b). With this in mind, we explored data from the TARGET-AML[71] project, which focuses on childhood cancers (Fig. 6c). Using TARGET data, we calculated hazard ratios using univariate Cox proportional-hazards modeling with continuous mRNA expression values for our genes of interest as independent variables. We observed that 4/7 FAS genes, GPAT4, CHP1, PCGF1, and GPI, show significant, negative hazard ratios (HR), consistent with a tumor suppressor signature (Fig. 6d), and that no other gene from our set shows a negative HR. Indeed, when stratifying patients from the TARGET cohort with high

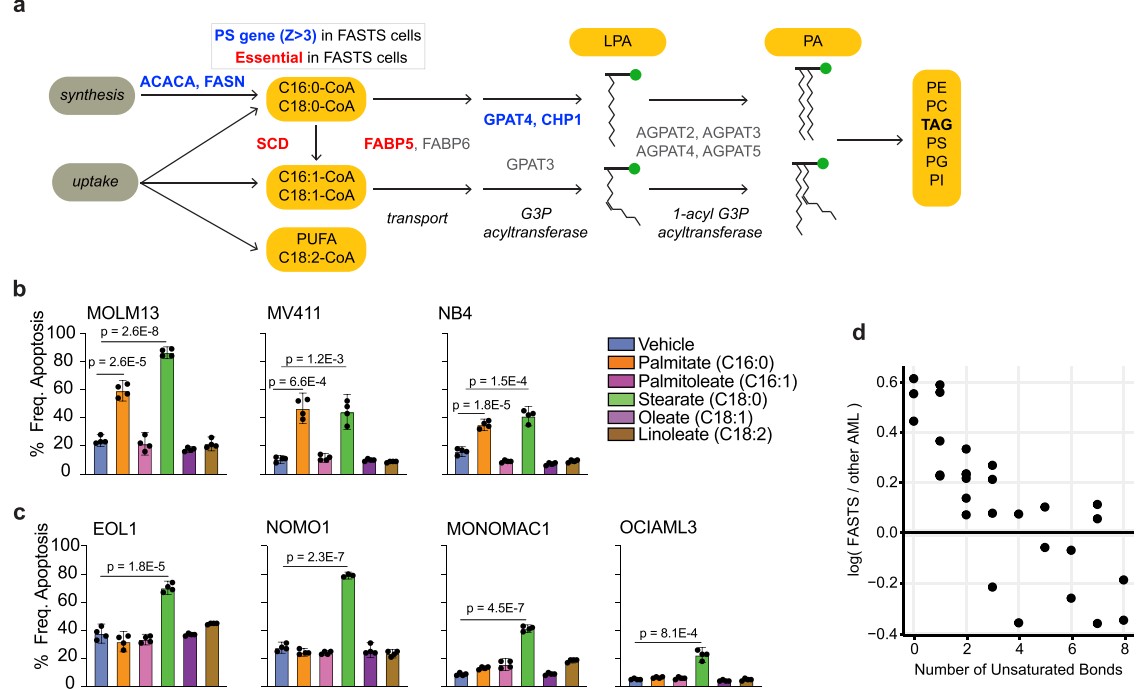

**Fig. 5 FASTS cells are sensitive to saturated FA. a** Schematic of the fatty acid/glycerolipid synthesis pathway. Blue, PSG in FASTS cells. Red, essential genes. Pathway analysis suggests saturated fatty acids are a critical node. **b** Apoptosis of FASTS cells in response to media supplemented with 200 μM fatty acids. All three cell lines show marked sensitivity to palmitate. Plotted bars in (**b**) show mean and 95% confidence interval (CI) of apoptosis %. *P* values in (**b**) and (**c**) represent one-sided unpaired *t* test comparisons. Each bar plot demonstrates four replicate samples for each condition and cell line tested. Data for (**b**) and (**c**) can be found in the source data file. **d** Triacylglycerol (TAG) species metabolite differences. The *x* axis represents the median difference of log10 normalized peak area of the metabolite in FASTS cells vs. all other AML cells. The *y* axis represents the number of saturated bonds present. Each dot represents a unique metabolite.

expression of *GPAT4, CHP1, PCGF1*, and *GPI* (Fig. 6e), we observe significantly improved survival (*P* value = 0.001, Fig. 6f). These findings are not replicated for *GPAT4, CHP1*, and *GPI* in the TCGA[72] or OHSU[73] tumor genomics datasets, possibly because they sample older cohorts (Polycomb group subunit *PCGF1* is observed to have a HR < 1 within the OHSU cohort, Supplementary Fig. 8a). However, age is not generally associated with the expression of genes in the FAS cluster in either cell lines or tumor samples (Supplementary Fig. 8).

## Discussion

CRISPR screens have had a profound impact on cancer functional genomics. While research has been mainly focused on essential gene phenotypes, there is still much clinically relevant biology that can be uncovered by examining other phenotypes from a genetic screen. We establish a methodology that can reliably identify the proliferation-suppressor phenotype from whole-genome CRISPR knockout genetic screens. Here, we present a systematic study of this phenotype in the more than 1,000 published screens[8,10,11,13,48].

The activity of proliferation-suppressor genes is inherently context-dependent, rendering global classification difficult. As with context-dependent essential genes, the strongest signal is attained when comparing knockout phenotype with underlying mutation state. For example, wild-type and mutant alleles of classic tumor suppressor examples *TP53* and *PTEN* are present in large numbers of cell lines, enabling relatively easy discrimination of PS behavior in wild-type backgrounds, but most mutations are much more rare, reducing statistical power. Our model-based approach enables the discovery of PS phenotype as an outlier from null-phenotype knockouts. Using this approach, we recover

COSMIC-annotated TSGs exhibiting the PS phenotype when wild-type alleles are expressed at nominal levels.

Co-occurrence of proliferation suppressors follows the principles of modular biology, with genes in the same pathway acting as proliferation suppressors in the same cell lines. We observe background-specific putative tumor suppressor activity for the PTEN pathway, P53 regulation, mTOR signaling, chromatin remodeling, and others. The co-occurrence network also reveals a module associated with glycerolipid biosynthesis, which exhibits the PS phenotype in a subset of AML cells. Analysis of the rewiring of the lipid metabolism coessentiality network in AML cells corroborated this discovery and led us to define the fatty acid synthesis/tumor suppressor (FASTS) phenotype in four AML cell lines. A survey of genetic interactions, using the enCas12a multiplex knockout platform, showed major network rewiring between FASTS and other AML cells and revealed strong genetic interactions in FASTS cells with *GPAT4*, a key enzyme in the processing of saturated fatty acids, and its regulator *CHP1*. Collectively these observations suggest that FASTS cells are near some critical threshold for saturated fatty acid carrying capacity, which we validated biochemically by treatment with fatty acids and bioinformatically through analysis of CCLE metabolomic profiles.

Confirming the clinical relevance of an in vitro phenotype can be difficult. No obvious mutation segregates FASTS cells from other AML cells, and with only four cell lines showing the FASTS phenotype, we lack the statistical power to discover associations in an unbiased way. However, by narrowing our search to strong hits from the differential network analyses, we found a significant survival advantage in a roughly age-matched cohort for *GPAT4* and *CHP1* overexpression. This finding points to a tumor

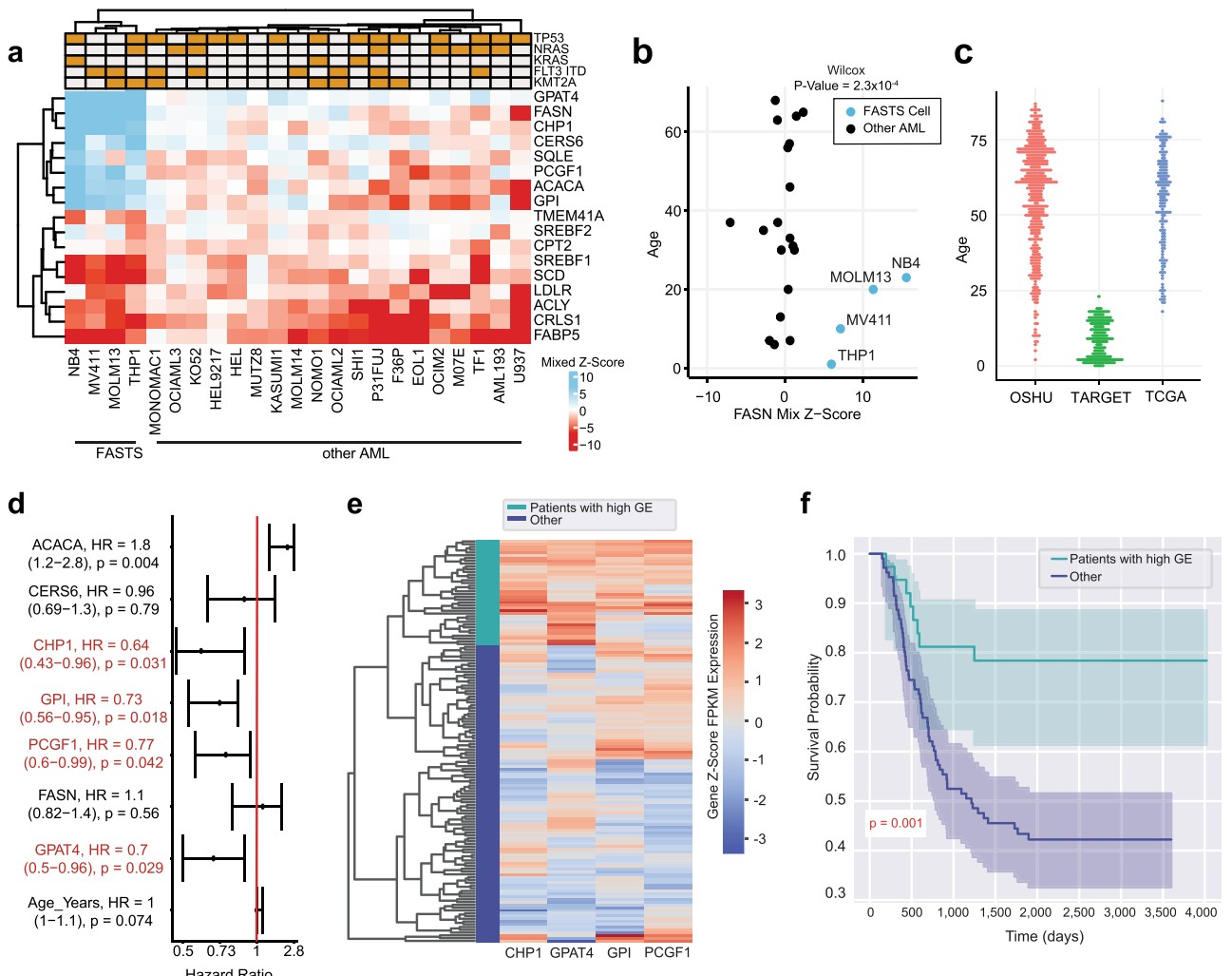

**Fig. 6 Prognostic signature of FAS module. a** Heatmap of mixed Z-scores for genes implicated in the genetic interaction network. Top, common AML lesions. **b** Mixed Z-score of FASN in AML cell lines vs. age of patient from which cell lines were derived. Blue, FASTS cells. **c** Age distribution of AML patients in three public tumor genomics cohorts. Red denotes OSHU, green denotes TARGET, and blue represents TCGA. **d** Hazard ratios (95% CI; univariate Cox proportional-hazards test) for expression of genes in (**a**), using genomics and survival data from TARGET. The dot represents the HR value indicated on the right. The 95% CI range for each point is indicated on the left. Genes indicated in red represent genes with significant HR less than 1 and are used for analysis in (**e**). **e** Hierarchical clustering of gene expression in TARGET, using the four genes with negative HR. Green, high expression cluster. Blue, others. **f** Kaplan–Meier survival analysis of AML patients in TARGET, comparing patients in high expression cluster vs. others. Shaded regions represent 95% CI of the curve. The number of independent patients analyzed in (**d**–**f**) = 145.

suppressor signature for our PSG module, though significant further study is necessary to determine whether this gene expression signature confers a similar in vivo metabolic rewiring and sensitivity to saturated lipids.

The combination of genetic, biochemical, and clinical support for the discovery of a tumor suppressor module has several implications. First, it provides a clinical signature that warrants further research as a prognostic marker as well as a potential therapeutic target. Second, it demonstrates the power of differential network analysis, and in particular differential genetic interaction networks, to dissect the rewiring of molecular pathways from modular phenotypes. Finally, it suggests that there still may be much to learn from data-driven analyses of large-scale screen data, beyond the low-hanging fruit of lesion/vulnerability associations.

## Methods

**Functions and packages related to data analysis**. Mixed Z-scoring, analysis using scoring metric, co-occurrence network, and survival analysis was conducted

in R version 4.0.4[74,75]. dPCC correlation analysis, including empirical calculations, was conducted in Python 3.8.2[76], using the packages SciPy[77], NumPy[78], Matplotlib[79], and pandas[80].

R packages tidyverse[81], data.table[82], and knitr[83–85] were used for figure generation, data manipulation, and general R functions; mixtools[86], permute[87], and PRROC[88,89] were used for data simulations present in figures and evaluation; biomaRt[90,91], and org.Hs.eg.db[92] were used in integrating data types; cowplot[93], ggbeeswarm[94], annotate[95], RColorBrewer[96], ComplexHeatmap[97], gplots[98], ggpubr[99], grid[75], circlize[100], ggthemes[101], ggExtra[102], patchwork[103], and ggplot2[104], were used for figure esthetics and generation. R packages survival[105,106] and survminer[107] were used for survival analysis and figure generation. Analysis related to Kaplan–Meier and patient stratification was done in python version 3.8.5[108] using the packages pandas[80], numpy[78], and scipy[77] for statistical functions and data manipulation, seaborn[109], plotly[110], and matplotlib[79] for figure esthetics and generation, and lifelines[111] for both statistical analysis and figure generation.

Analysis of enCas12a multiplex genetic screens was conducted in R 4.0.0[75] and Python 3.8.3[112]. Code for this analysis is available at https://github.com/PeterDeWeirdt/FASTS. R packages tidyverse[81] and tidygraph[113] were used for data manipulation and ggraph[114] was used for graph visualization. Python packages SciPy[77], NumPy[78], Matplotlib[79], pandas[80], statsmodels[115], plotnine[116] were used for analysis and visualization. The Custom package gnt[117] was used to calculate genetic interaction scores and gpplot[118] was used to generate point density plots.

**Processing DepMap screen and CCLE genomics data**. Raw read count data and a map of guide RNAs were downloaded from the DepMap database (www.depmap.org)[10,48] and Project Score database (https://depmap.sanger.ac.uk/)[13]. Avana data version 2020q4[49] was used for this analysis. To avoid genetic interaction effects, we discarded sgRNAs targeting multiple protein-coding genes annotated as public or update pending in The Consensus Coding Sequence (CCDS, release 22)[119]. Gene names in the guide RNA maps of Avana and Project Score were updated using human gene information obtained from ncbi ftp. Then, read count data for each replicate was passed through CRISPRcleanR[120] with location information of sgRNAs for the Avana CRISPR library based on GENCODE[121] to correct depletion effects caused by copy-number amplification. Following this correction, each guide's log$_2$ fold change was calculated. For Project Score data, we used only the gene location information of KY library v1.0 which is built in CRISPRcleanR. Normalized TPM RNA-seq data, copy-number data, and mutation annotations for CCLE[69] cells were also downloaded from DepMap. Ensembl gene id in RNA-seq data was converted to gene symbol using cross-reference downloaded from Emsembl Biomart[122].

**Mixed Z-score metric**. Mixed Z-score metric was generated using R version 4.0.4 base stat packages[75] and the mixtools[86] normalmixEM function. To calculate the mixed Z-score, individual guide log$_2$ fold changes for each cell line were passed through the default settings of the normalmixEM function to fit two distinction normal distributions. Of the 808 cell lines passed through this analysis, 805 cell lines were able to converge with two distinction normal distribution following 1000 iterations. The calculated mean and standard deviation of the higher (more positive) distribution were recorded. Along with the uncorrected original gene log$_2$ fold change, was used to calculate the corresponding mixed Z-score. The original and mixed Z-score equation is as follows:

$$\text{Mixed Gene Z} - \text{Score} = \frac{x - \mu_{\text{high}}}{\sigma_{\text{high}}} \quad (1)$$

Where $x$ is the original gene log$_2$ fold change, $\mu_{\text{high}}$ is the average of the more positive fitted distribution, and $\sigma_{\text{high}}$ is the standard deviation of the more positive fitted distribution. This metric was calculated for the DepMap 2020q4[49] screen set, and the Sanger's DepMap[13] screen set for Supplementary Fig. 3. Visualization of the mixed Z-score for the Broad's and Sanger DepMap screen sets can be seen at the PICKLES[123] database: https://pickles.hart-lab.org/.

**Comparisons of fitness-scoring metrics**. The following describes our comparative analysis of screening algorithms observed in Supplementary Fig. 1. JACKS[43] and BAGEL[41,42,124], software was downloaded from their corresponding GitHub official distribution sites: https://github.com/felicityallen/JACKS, and https://github.com/hart-lab/bagel. We ran JACKS and BAGEL with raw fold-change data of DepMap 2020q4 version[49], gene guide map, and replicate information. We obtained DepMap 2020q4 CERES scores from "dependency_score.csv" downloaded from DepMap depository. Ranking was performed per screen and based on mean log$_2$ fold-change values per gene.

We used the cancer gene census (CGC) list from COSMIC[45,46] to compare fitness methods that can detect proliferation-suppressor activity. Tumor suppressor genes (TSGs) from CGC represent a gene set of well-known proliferation suppressors. We separated the CGC gene list in two gene sets, genes with any tumor suppressor role in cancer representing true positive proliferation-suppressor observations, and genes with any oncogene role in cancer representing false positives. In addition, we added reference nonessential genes[7,47] to the false-positive list as these genes are not expected to demonstrate any phenotype. With these compiled lists, we evaluated each metric's fitness scores, to see which metric would best separate the true and false-positive gene lists. The R package PRROC was used for fitness-scoring evaluation[88,89].

**Direct proliferation-suppressor comparisons of Avana and Sanger screen datasets**. The CRISPRcleanR[120] corrected fold-change Sanger screen set[13] was pushed through identical pipelines used to calculate the mixed Z-score metric. Quality analysis of the mixed Z-score metric for both datasets was pushed using identical gene sets described in the "Comparisons of Fitness Scoring Metrics" section. This analysis was restricted to only overlapping cell lines, 186 total, in both datasets. Cell lines were matched using the Cell Model Passports database[125].

The fitness enhancement introduced by PSG knockout, relatively weak compared to severe defects from essential gene knockout, often precludes detection in a shorter experiment. In the example F5 cell line (Fig. 1a), a 2.5-fold change over a 21-day time course corresponds to a fitness increase of only ~12% for rapidly growing cells, or a doubling time decrease from 24 to 21 h. In a 14-day experiment, this increased proliferation rate would result in an observed log-fold change of only ~1.7, within the expected noise from genes with no knockout phenotype. This is explained in detail as follows:

Theoretical fold-change and growth rate quantification: To assess hypothetical differences of proliferation-suppressor fitness-scoring metrics based on standard sampling times of screen collection taken from the Sanger and Avana databases[10,11,13,48], we calculated theoretical cell population differences of wild-type and knocked-out proliferation-suppressor cell lines. The following Eq. (2) can

be used to calculate cell populations based on doubling rate per day:

$$X_f = X_i * 2^{k*t} \quad (2)$$

In this formula, $X_f$ is the final population number of cells, $X_i$ is the initial population of cells, $k$ is the doubling time of the cells (in days), and $t$ is time in days. In order to compare cells, we can assume that these formulas are consistent with both wild-type cells and knocked-out proliferation-suppressor cells. With, knocked-out proliferation-suppressor cells the assumption is that these cells would grow faster compared to wild-type conditions and thus $k_{ps} > k_{wt}$, where $k_{ps}$ is the growth rate for proliferation-suppressor knocked-out cells, and $k_{wt}$ is the growth rate of wild-type cells. These two independent growth rates are related as:

$$k_{ps} = k_{wt} + \Delta k \quad (3)$$

$\Delta k$ represents the change in growth rate resulting from genetic knockout and is assumed to be positive. The growth rate equation for wild-type and proliferation-suppressor cells is thus:

$$X_{wt} = X_i * 2^{k_{wt}*t}, \; X_{ps} = X_i * 2^{(k_{wt} + \Delta k)*t} \quad (4)$$

We then solved for $\Delta k$, with Log$_2$($X_{ps}/X_{wt}$) as Log$_2$(FC), representing the fold-change difference between the cell populations at time $t$:

$$\text{Log}_2\text{FC} = \text{Log}_2\left(\frac{X_{ps}}{X_{wt}}\right) \quad (5)$$

$$\text{Log}_2\text{FC} = \text{Log}_2\left(\frac{X_i * 2^{(k_{wt} + \Delta k)*t}}{X_i * 2^{k_{wt}*t}}\right) \quad (6)$$

$$\text{Log}_2\text{FC} = \text{Log}_2\left(\frac{2^{(k_{wt} + \Delta k)*t}}{2^{k_{wt}*t}}\right) \quad (7)$$

$$\text{Log}_2\text{FC} = ((k_{wt} + \Delta k) * t) - (k_{wt} * t) \quad (8)$$

$$\text{Log}_2\text{FC}/t = k_{wt} + \Delta k - k_{wt} \quad (9)$$

$$\text{Log}_2 FC/t = \Delta k \quad (10)$$

For a representative Log$_2$(FC) of 2.5, which represents a sizable gain in fitness from a knocked-out proliferation-suppressor, and $t = 21$ days, representing the time in which the Avana screens were sampled, we calculated $\Delta k$:

$$\Delta k = \frac{2.5}{21} = 0.12 \quad (11)$$

Using the calculated $\Delta k$ at 0.12, we can calculate the hypothetical Log$_2$(FC) that would be expected at $t = 14$ days, representing the time in which the Sanger screens were sampled:

$$\text{Log}_2\text{FC} = \Delta k * t \quad (12)$$

$$\text{Log}_2\text{FC} = 0.12 * 14 = 1.7 \quad (13)$$

The resulting theoretical measurements demonstrate that $\Delta k$ can be identical between two samples, however, the time in which the sample was taken will influence the ratio between the two measured cell populations. Taken together, this demonstrates that samples at shorter time points will demonstrate smaller quantified population size differences between wild-type and proliferation-suppressor knocked-out cells compared to samples taken at longer time points.

**Proliferation-suppressor co-occurrence network**. The co-occurrence network was developed based on FDR-corrected P values from Fisher exact tests of all gene-by-gene comparisons that were identified as a proliferation suppressor more than once (584 genes total). Parallel processing, Fisher's exact test, and Benjamini & Hochberg FDR P value adjustment were done using base R stat packages[75]. Figure 2a was created with heatmap.2 function from the R gplots[98] package, with the dendrogram created through base R[75] functions of Euclidean distance, and complete agglomeration methods clustering of the Fisher's exact test score between gene pairs. Smaller heatmaps displayed in Fig. 2c were made using the R ComplexHeatmap library[97]. Network visualization was completed using Cytoscape[126].

Network creation followed the corresponding steps; (1) identify all proliferation-suppressor observations at a 10% FDR threshold (Z > = 3.83). (2) Filter for gene proliferation-suppressor observations that occurred at least 2 or more times, selecting for a total of 584 out of 18,111 genes available (3.2% total available genes); (3) Create a binary (1 = proliferation suppressor, 0 = not proliferation suppressor) matrix of all 584 genes in all cell lines; (4) Conducted Fisher's exact test of every possible 2 × 2 contingency table of the 584 selected genes ($n = 170{,}236$ tests); and (5) Adjust the corresponding P values to FDR values, using a cutoff of 0.001 (0.1% FDR) to define edges. By assessing gene edges through Fisher exact tests, we observe gene associations that are based on the relative proportion of co-occurrences between two genes.

**Proliferation-suppressor network enrichment**. To test network enrichment of observed edges (Supplementary Fig. 4a), we took 10,000 random samples of 462 (total number of edges in the co-occurrence network) gene pairs from the 170,236 available all by all gene pair Fisher's exact test set. We then compared each sample to see the frequency of gene pairs observed to have some interaction within HumanNet[61], excluding genetic interactions observed solely in the coessentiality network component[21] (generated from the same data) to prevent circularity. In addition, we compared our selected mixed Z-score cutoff against other various Z-score cutoffs to ensure that we observed appropriate edge representation from HumanNet (Supplementary Fig. 4b). Networks were made using identical pipelines and Fisher's exact test set cutoffs with Z-score cutoffs between 3 and 8 at 0.2 increments.

**Differential Pearson correlation coefficient analysis**. Differential Pearson correlation coefficient (dPCC) analysis was conducted to identify genetic fitness distinctions between AML cells and all other cells (Fig. 3). Initial correlations (Fig. 3a) of FAS cluster genes, *PCGF1*, *CERS6*, *GPI*, *FASN*, *CHP1*, *GPAT4*, and *ACACA* were calculated with R version 4.0.4 base stat packages[75] and plotted in ggplot2[104].

Following this observation, a follow-up dPCC analysis was conducted on the FASTS cluster genes to assess dPCC quality. Cell line screens with low quality (Cohen's $D < 2.5$ or recall of known core-essential genes <60%) were excluded, leaving 659 cell lines. Following this filtering step, two gene-by-gene correlation matrices were calculated. The first correlation matrix calculated all gene-by-gene pairs in only the available AML cell lines ($n = 17$). The second matrix calculated all gene-by-gene pairs in the remaining 642 cell lines. The dPCC matrix is therefore the AML correlation matrix minus the non-AML correlation matrix.

Each gene pair has a unique joint distribution of mixed Z-scores; thus, the significance of each dPCC score must be calculated individually. To do this, we generated null distributions for dPCC for each gene pair. We took random selections without replacement of 17 cell lines (matching the $n$ of AML cells), calculated all gene-by-gene pairwise correlations within this selection and within the remainder, and calculated dPCC. We repeated this sampling and calculation 1000 times to generate a unique null distribution of dPCC for each gene pair and calculated an appropriate $P$ value for the observed dPCC above (right-tailed for positive dPCC, left-tailed for negative dPCC).

Genes which showed significant knockout phenotype ($|$mixed $Z| > 5$) and AML-specific change in correlation (dPCC $P < 0.001$) with a gene in the connected clique in the co-occurrence cluster (*CHP1*, *GPAT4*, *ACACA*, *FASN*, *GPI*, *CERS6*, *PCGF1*) were selected for further analysis (Fig. 3e). Figure 3e was made using the R ComplexHeatmap library[97]. Figure 3c, d plots were made using the Python package Matplotlib[79].

**Cell culture for genetic screens**. MOLM13 and NOMO1 cells screened with the Cas12a-mediated genetic interaction library at the Broad Institute were obtained from the Cancer Cell Line Encyclopedia.

All cell lines were routinely tested for mycoplasma contamination and were maintained without antibiotics except during screens, when the media was supplemented with 1% penicillin/streptomycin. Cell lines were kept in a 37 °C humidity-controlled incubator with 5.0% carbon dioxide and were maintained in exponential phase growth by passaging every 2−3 days. The following media conditions and doses of polybrene, puromycin, and blasticidin, respectively, were used:

MOLM13: RPMI + 10% FBS; 8 μg mL$^{-1}$; 4 μg mL$^{-1}$; 8 μg mL$^{-1}$
NOMO1: RPMI + 10% FBS; 8 μg mL$^{-1}$; 1 μg mL$^{-1}$; 8 μg mL$^{-1}$

**Pooled screens**. Cell lines stably expressing enCas12a (pRDA_174, Addgene 136476) were transduced with guides cloned into the pRDA_052 vector (Addgene 136474) in two cell culture replicates at a low MOI (~0.5). Transductions were performed with enough cells to achieve a representation of at least 750 cells per guide construct per replicate, taking into account a 30–50% transduction efficiency. Throughout the screen, cells were split at a density to maintain a representation of at least 1000 cells per guide construct, and cell counts were taken at each passage to monitor growth. Puromycin selection was added 2 days post-transduction and was maintained for 5 days. Fourteen days and 21 days after transduction, cells were pelleted by centrifugation, resuspended in PBS, and frozen promptly for genomic DNA isolation.

**Genomic DNA isolation and PCR**. Genomic DNA (gDNA) was isolated using the KingFisher Flex Purification System with the Mag-Bind® Blood & Tissue DNA HDQ Kit (Omega Bio-Tek #M6399-01) as per the manufacturer's instructions. The gDNA concentrations were quantitated by Qubit. For PCR amplification, gDNA was divided into 100 μL reactions such that each well had at most 10 μg of gDNA. Per 96-well plate, a master mix consisted of 144 μL of 50× Titanium Taq DNA Polymerase (Takara), 960 μL of 10x Titanium Taq buffer, 768 μL of dNTP (stock at 2.5 mM) provided with the enzyme, 48 μL of P5 stagger primer mix (stock at 100 μM concentration), 480 μL of DMSO, and 1.44 mL water. Each well consisted of 50 μL of gDNA plus water, 40 μL of PCR master mix, and 10 μL of a uniquely barcoded P7 primer (stock at 5 μM concentration).

PCR cycling conditions: an initial 1 min at 95 °C; followed by 30 s at 94 °C, 30 s at 53 °C, 30 s at 72 °C, for 28 cycles; and a final 10 min extension at 72 °C. PCR primers were synthesized at Integrated DNA Technologies (IDT). PCR products were purified with Agencourt AMPure XP SPRI beads according to the manufacturer's instructions (Beckman Coulter, A63880).

Samples were sequenced on a HiSeq2500 Rapid Run flowcell (Illumina) with a custom primer of sequence: 5'-CTTGTGGAAAGGACGAAACACCGGTAATTTCTACTCTTGTAGAT. The first nucleotide sequenced with the primer is the first nucleotide of the guide RNA, which will contain a mix of all four nucleotides, and thus staggered primers are not required to maintain diversity when using this approach. Reads were counted by alignment to a reference file of all possible guide RNAs present in the library. The read was then assigned to a condition (e.g., a well on the PCR plate) on the basis of the 8 nt index included in the P7 primer.

**Scoring genetic interactions**. To score genetic interactions we used a custom python package, gnt[117], available on the python package index. We use log-fold changes (LFCs) as inputs to the scoring pipeline. We define $y_{ij}$ as the observed LFC of a guide pair $i, j$, and $\hat{y}_{ij}$ as this pair's expected LFC. We then calculate the residual $y_{ij} - \hat{y}_{ij}$ to generate an interaction score. To define expected LFCs, $\hat{y}_{ij}$ we fit a linear regression for each guide, $i$, saying

$$\hat{y}_i = m_i \cdot x + b_i, \tag{14}$$

where $x$ is the LFC of each guide individually and $m_i$ and $b_i$ are the fit slope and intercept for guide $i$ (Supplementary Fig. 6b). We refer to $i$ as the anchor guide and its pairs as target guides. We then Z-score residuals within each anchor guide. This approach is similar to the one taken by Horlbeck et al.[33].

To aggregate interaction scores at the gene level, we sum the Z-scored residuals, $z_{ij}$, for all constructs $i, j$ targeting the gene pair $I, J$, fixing $I$ as the anchor gene, and divide by the square root of the number of constructs targeting $I, J$. We repeat this calculation, fixing $J$ as the anchor gene. We sum scores for both of these orientations and divide by $\sqrt{2}$ to arrive at a gene-level Z-score.

**Cell culture for fatty acid response**. Human cancer cell lines used at MD Anderson were obtained as follows: EOL1, MONOMAC1, NB4, OCIAML3 (DSMZ, #ACC-386 #ACC-252 #ACC-207 #ACC-582); MOLM13 and NOMO1 (Fisher, #NC0442994 #NC1515509); MV411 (ATCC #CRL-9591). Identities were confirmed upon receipt and prior to experiments by STR typing (MDACC Characterized Cell Line Core). The absence of mycoplasma was confirmed monthly (Invivogen #rep-pt1). All cell lines were grown at 37 °C in 5% CO$_2$ in low attachment flasks (Greiner) and maintained at less than 1 M cells ml$^{-1}$. All but one line were cultured in RPMI-1640 with 25 mM HEPES ((Sigma #R5886) supplemented with 10% FBS (Sigma # F0926), 2 mM Glutamax (Gibco #35050061), 1 mM sodium pyruvate (Gibco #11360070), 10,000 units ml$^{-1}$ penicillin (Gibco #15140122), 10 mg ml$^{-1}$ streptomycin (Gibco #15140122), and 100 μg ml$^{-1}$ Normocin (Invivogen #ANTNR2). Complete medium was additionally supplemented with 0.1 mM nonessential amino acids (Gibco #11140050) for MONOMAC1.

**Fatty acid solutions**. Fatty acid chemicals were purchased from Sigma (St. Louis, MO). Solutions were prepared according to Luo et al.[127] following best practices[128]. Fatty acid stock solutions were prepared in 100% ethanol at 50 mM for stearic acid or 200 mM for the rest. Fatty acid-free bovine serum albumin (FAF-BSA) was dissolved in tissue culture grade (pyrogen-free) water at 1.5 mM (10% w/v), filtered using 0.1 μm PES vacuum unit (Corning) and aliquoted for storage at −20 °C. Ethanol stock solutions were diluted to 4 mM in FAF-BSA (molar ratio 2.7:1) and mixed gently at room temperature for 2 h to facilitate conjugation. A vehicle control was prepared by diluting 100% ethanol in FAF-BSA to match the ethanol concentration in the 4 mM stearic acid solution. Vehicle or 4 mM solutions were aliquoted and stored at −80 °C for up to 3 months. After thawing, aliquots were diluted 1:10 with complete medium to 400 μM, stored at 4 °C and used within 1 week.

**Apoptosis assay**. Cells were seeded 24 h prior to treatment in 500 μL complete medium in 24-well low attachment plates (Greiner) at 250,000 cells well$^{-1}$. Quadruplicate wells received 500 μL FA working solution (400 μM) or vehicle (BSA +EtOH). Cells were treated at 200 μM for 48 hr. Treated cells were transferred to a deep 96-well plate and medium was discarded after centrifugation at 500×$g$ for 5 min. Cells were washed once with 1000 μL D-PBS (Sigma #D8537). Next, cells were resuspended in 300 μL binding buffer containing annexin-FITC (BD Biosciences #BD556547) and propidium iodide (Invitrogen #P3566) according to the manufacturer's protocol (BD Biosciences) and transferred to a shallow 96-well V-bottom plate (Corning). After staining for 15 min at room temperature in the dark, cells were washed once with 300 μL binding buffer and finally resuspended in 100 μL binding buffer. Unstained and single-stain controls were prepared for every cell line in a separate plate. Gates were adjusted such that 99% of unstained singlets fell below each threshold. See Supplementary Fig. 9 for the complete gating strategy. Flow cytometry data were collected using a FACSCelesta analyzer equipped with an autosampler (BD Biosciences) and analyzed using FlowJo 10.5.3. The results shown are representative of three independent experiments conducted

with sequential passages of each cell line. Statistical tests shown in Fig. 5b, c were one-sided unpaired *t* tests of the apoptosis percentages and were calculated using base R statistic[75] functions.

**Metabolomics analysis**. This section describes the methods used within Fig. 5d and Supplementary Fig. 7. Metabolomics data acquired from Supplementary Table 1 of Li et al.[70]. For analysis, normalized data ("1-clean data") and coefficient of variation for each metabolite ("1-CV") was used. Normalized data were filtered to select only AML cells that were present in the Avana 2020q4[49] screen set. Following filtering, the median of species present was taken, grouped by whether the measurement was from a FASTS AML or other AML cell line. The difference in median, representing the log ratio, was taken for each metabolite. Metabolites that had differences in medians less than the coefficient of variation were omitted from the plots. Acyl group and the number of unsaturated bonds were obtained directly from the provided nomenclature.

**AML patient survival analysis**. This section describes the methods used within Fig. 6 and Supplementary Figs. 8 and 10. Genes chosen for analysis were all genes shown to have an interaction with *ACACA* in Fig. 4h and *FASN*. Gene annotations noted in the Fig. 6a heatmap include any nonsilent mutation, copy-number loss for *TP53* and *KMT2A*, and copy-number gain for *KRAS*, *NRAS*, and *FLT3*. *FLT3-ITD* annotations were included in the *FLT3* annotation row bar. Mutation annotations come from CCLE[69], copy-number calls come from the cBioPortal[129,130] database, and *FLT-ITD* annotations come from the DSMZ catalogue[131].

TARGET-AML[71] data including age, genetic expression (HTseq FPKM UQ), time to event, and survival event outcomes, and TCGA[72] patient ages and genetic expression were downloaded directly from the Xena[132] database. The OHSU BeatAML[73] age data was directly downloaded from the Vizome database, and genetic expression data were taken from the original publication. Age of patient-derived cell lines was obtained from the Cellosaurus database[133]. Hazard ratios calculated from Cox proportional-hazards modeling were done using the R survival[105,106] package. Patient clustering stratification was done with clustering functions from the scipy package[77], using Euclidean clustering and complete linkage settings. This output heatmap of TARGET-AML patients (Fig. 6e) was created using functions from the seaborn[109] package. We identified the patient cluster containing the highest overall expression of *CHP1*, *GPAT4*, *GPI*, *PCGF1* from the heatmap using the fcluster function from scipy[77]. Figure 6f demonstrates the resulting survival comparison of the two patient clusters and was created with functions from the lifelines[111] package, specifically, KaplanMeierFitter (alpha = 0.05, default) function for the Kaplan–Meier curve, and the *P* value reflecting the calculated log-rank test of the two curves.

*P* values related to Schoenfeld tests calculated internally by the survminer package. For TARGET data analysis, patient expression profiles were chosen from primary tumor samples, filtering out samples from recurrent patients (42 such cases). Patient stratification is conducted based on stratifying patient groups into lower genetic expression (patients with genetic expression below the 75th percentile, *n* = 108 independent patients), and higher genetic expression (patients with 75th percentile and above, *n* = 37 independent patients). Computed hazard ratios for all tested genes within the TARGET cohort all passed the Cox proportion hazards assumption (Supplementary Fig. 10) by failing to reject the Schoenfeld test null hypothesis.

**Reporting summary**. Further information on research design is available in the Nature Research Reporting Summary linked to this article.

## Data availability

Genetic Interaction (enCas12a) data pertaining to Fig. 4 and Supplemental Fig. 6 can be found at https://github.com/PeterDeWeirdt/FASTS. Figure 5b–c data can be found within the source data file. Cytoscape[126] network files of PSG network (Fig. 2 and Supplemental Fig. 4) can be found at https://doi.org/10.6084/m9.figshare.16746052.v1. Relevant data for figures, including gene Mix Z-score evaluation, fisher edge calculations, dPCC scoring metrics, and other screen metric comparisons, can be found at https://doi.org/10.6084/m9.figshare.16746040.v1. External data used in this study include the screening set coming from the Avana 2020q4[10,48,49] release, and CCLE[69] genetic expression, mutation, and copy-number data that can be found at www.depmap.org; screening data used from Project Score[13] that can be found at https://depmap.sanger.ac.uk/; Cell Model Passports[125] data were used in screening data comparison and can be found at https://cellmodelpassports.sanger.ac.uk/; the cancer gene census[45,46] used to define oncogenes and tumor suppressors that can be found at https://cancer.sanger.ac.uk/census; absolute gene copy-number values from cell lines obtained the cBioPortal database[130] at https://www.cbioportal.org/; HumanNet[61] data used for network comparisons can be found at https://www.inetbio.org/humannet/; the Xena database[132] was used in acquiring specific data related to the TCGA LAML, TARGET AML, and BeatAML datasets and can be found at https://xenabrowser.net/; and additional BeatAML analysis was taken directly from Tyner et al.[73] publication. The results published here are in part based upon data generated by the Therapeutically Applicable Research to Generate Effective Treatments (TARGET) initiative, phs000218, managed by the NCI. The data used for this analysis are available at dbGaP Study

Accession: phs000465.v19.p8. Information about TARGET can be found at http://ocg.cancer.gov/programs/target. Source data are provided with this paper.

## Code availability

Genetic Interaction (enCas12a) code notebooks pertaining to Fig. 4 and Supplemental Fig. 6 can be found at https://github.com/PeterDeWeirdt/FASTS. Code pertaining to all figures except for Fig. 4, Supplemental Fig. 6, and 9 is available at: https://doi.org/10.6084/m9.figshare.16786063. Additional analysis code (primarily co-occurrence network, mixed Z-score metrics, dPCC correlation, and clinical analysis) is available at https://doi.org/10.6084/m9.figshare.16786078.v1.

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

## Acknowledgements

This research was performed in partial fulfillment of the requirements for the PhD degree from The University of Texas MD Anderson Cancer Center UTHealth Graduate School of Biomedical Sciences; The University of Texas MD Anderson Cancer Center, Houston, Texas 77030. W.F.L., M.Mc., M.Mo. and T.H. were supported by NIGMS grant R35GM130119. MC is supported by a Kopchick fellowship and Pauline Altman-Goldstein Foundation Discovery Fellowship. E.K. is supported by a grant from the Prostate Cancer Foundation. M.D. is supported by a Schissler Foundation fellowship. T.H. is a CPRIT Scholar in Cancer Research (RR160032), and is additionally supported by MD Anderson Cancer Center Support Grant P30 CA016672. W.F.L. is supported by the American Legion Auxiliary Fellowship in Cancer Research. This work was supported by the Andrew Sabin Family Foundation Fellowship (T.H.). Flow cytometry was performed at MDACC's Advanced Cytometry & Sorting Facility supported by the NCI Cancer Center Support Grant P30CA16672.

## Author contributions

W.F.L. performed all PS discovery analyses. M.F., A.G. and A.S. performed genetic interaction screens; and P.D. and M.C. performed bioinformatic analysis. W.F.L., M.C., N.E.A., E.K. and M.D. performed all other bioinformatic analyses. M.Mo. and M.Mc. performed lipid profiling experiments. J.G.D. and T.H. supervised the research. W.F.L. and T.H. drafted the manuscript and all authors edited it.

## Competing interests

J.G.D. consults for Agios, Maze Therapeutics, Microsoft Research, and Pfizer; J.G.D. consults for and has equity in Tango Therapeutics. W.F.L. has equity in Kronos Bio Inc. The remaining authors declare no competing interests.
