## [Peer review file · Nature Communications]

REVIEWER COMMENTS

Reviewer #1 (Remarks to the Author): Expert in bioinformatics and CRISPR screening

This study seeks to develop a systematic approach to classify the suppressors of proliferation genes whose knockout provides a proliferation advantage in vitro. Whole genome CRISPR knockout genetic screens have had a profound effect on the cancer functional genomics and recently large functional genomic screens in cancer cell lines become available especially in the form of Cancer Dependency Map (DepMap). Although essential gene phenotypes are examined in extensive detail there is not really much research on the proliferation suppressor phenotype. Here in this paper the authors aim to develop a framework to examine this phenotype in more detail. Using this analysis framework a network of 103 genes in 22 modules are identified. One of these modules contain several genes from glycerolipid biosynthesis pathway and operates exclusively in a subset of AML lines. A Cas12a-mediated genetic interaction screen is used to confirm the underlying differences in the lipid biosynthesis genetic interaction network between two AML cell lines. Using this screen a novel genetic interaction is identified between GPAT4 and CHP1 which has a clinical relevance on the survival.

This is a really interesting study where proliferation suppressor genes in CRISPR knockout genetic screens are performed. They analysed all the cell lines from Broad DepMap set and identified tumour suppressor genes. From this general analysis the paper focused on a novel module that is associated with fatty acid and lipid biosynthesis pathways. From here on the paper explores a subset of AML cell lines and identified a genetic interaction that has a clinical relevance. I am quite supportive of the goals of this study in principle where you transform the results of a general analysis of CRISPR knockout genetic screens to clinically relevant conclusion. However the paper is quite difficult to follow where the objective of the study is changing constantly and jumping from one result to another without a conclusion. There are also some issues about the results and the conclusions of the paper.

The paper is overall is written quite well and the methods are explained in detail, but there are some typos and grammatical errors in the paper including the supplementary part. Also there are some issues with the Figure names in some places.

The proliferation suppressor genes (PSG) are identified using a data driven approach where the mean LFC of gRNAs targeting a gene are compared to a variance normalized null distribution. The null distribution is achieved by shuffling the labels of the guide level LFCs to calculate gene level mean fold change. This is repeated 1000 times. Although the method is not a novel approach, it is convincing. The results are compared to a set of known tumour suppressors. The change in the recall criteria is not justified convincingly. The study focuses on the tumour suppressors that identified in the analysis but there is no mention of the novel tumour suppressors that are identified in the analysis. In fact study focuses on 58 tumour suppressors that overlap with the COSMIC TSGs but there is no information about the other TSGs detected by this method. This study is applying this method on 563 cell lines and only focuses on known TSGs. All the mutation and gene expression analysis is performed on the 58 known TSGs. I think the analysis should be extended to all results of the shuffling method. Although gene expression and silent mutations are inspected there is no information on the copy number of these genes which might be interesting too.

Another issue with the study is the detailed analysis of the CRISPR screen dataset from Sanger. There is a detailed explanation of why the screen is not used. It seems that part is a little bit more detailed and may be shortened as the data is not used at the end. In the supplementary there is a really nice explanation about why it is not used. However in the supplementary it seems that it might be possible to adjust for the time difference and maybe the data might be integrated. It would be also interesting to check the PSGs that are identified by the shuffling method in this dataset and check if they cannot be detected at all.

The correlation network for the PSG is quite interesting and novel idea. However limiting the networks

to only PSG might be a little limiting. It would be interesting to include essential genes to the networks to see how it affects the connectivity. Another limitation is to use only the PSGs that are observed in two cell lines. This will make it hard to investigate rare PSGs. Overall co-occurrence PSG networks is an interesting and novel idea. It would be also interesting to compare these networks with the PPI networks or their shortest path distance over the PPI networks.

Although some information is given on these modules there is not a good characterization of the modules other than the fatty acid pathways. Is this due to the fact that they are not interesting or they are well known pathways or there are not enough genes in the modules. It would be interesting to look at the pathway enrichment of all the modules together.

The validation of the genetic interaction identified in the subset of AML cell lines using the Cas12a-mediated genetic interaction is quite exciting. However the method that are used for detecting the gene interaction scores is not very well described. In the study only two cell lines are used and it is not clear if there are any replicates. Also they have a 14 day and 21 day samples but the 14 day sample results are not mentioned in the results. It would confirm the difference in the previous section related to the time course of CRISPR knockout genetic screens. Also it would be more effective to run the screen on all the 4 AML cell lines that are FASTS and 4 non FASTS cell lines.

The clinical relevance part is quite convincing. It will require a little bit more inspection on the mutational signature that drive the phenotype. The gene expression observation is quite important clinically and important. It would be worth checking the same expression profile in other cancer types and tissues in order to understand why it is specific to a subset of AML.

There some small issues with the paper and the results section can be extended a little bit more. However it is quite novel paper in terms of the development of the study. The differential correlation of shuffled Z-scores can be applied for other cell line sub-groups. The paper might benefit from a general figure that shows the whole process from the initial DepMap analysis to the clinical analysis. Overall the paper is written well except some small issues which can be addressed by going over the text and the methods are described in detail. All the software used in the study as well as their versions are identified clearly. Although the github pages are not accessible it seems the scripts are there. The experimental protocols are described in detail in the supplementary and tables are included in the submission.

Reviewer #2 (Remarks to the Author): Expert in AML and metabolism

The manuscript by Lenoir and colleagues describes a new method for analyzing CRISPR screen data to identify genes that suppress cancer cell line growth. Utilizing this method, the authors identify that fatty acid synthesis acts as a tumor suppressor in a subset of AML cell lines. In a pediatric patient cohort elevated expression of genes involved in fatty acid metabolism were shown to be associated with a survival benefit. While the results are intriguing several approaches could be taken to strengthen the overall conclusions. Further, additional explanation of how this method could be used in future screens would be helpful (see comment 1).

1. The CRISPR screen analysis was not reproducible between the two datasets analyzed (Behan et al. and Avana) using the authors new analysis method. Dempster et al. Nature Communications, 2019 reported that these datasets had large agreement between them despite differences in the experimental design. The authors contribute the difference in their method to a shorter assay time. Do the authors think that their method is limited in use to longer-term screening approaches?
2. The authors caution the use of FASN inhibitors. Can the authors show FASN inhibitors increases cell proliferation and viability in the FASTS and not other AML cell lines?

3. The authors should show if the FASTS signature correlates with sensitivity to fatty acids in primary AML specimens.

4. Does the FASTS signature correlate with age in the different AML patient cohorts? Can the authors provide an explanation for differences observed in the adult vs pediatric patient cohorts?

Reviewer #3 (Remarks to the Author): Expert in AML and metabolism

This is a very interesting computational study based on identifying gene modules associated with fitness changes in CRISPR screens of AML cells. The authors have defined a very interesting pipeline for analyzing these data and making potential predictions about AML outcome. The biological validation of the computational approach is in its current form however not sufficient to rule out alternative hypotheses and support the authors' conclusions and the translational potential of the work.

In figure 6, the survival probability shown in figures E and F could be explained by at least two alternative hypotheses - 1) the leukemia is less 'proliferative' due to lipid signaling or 2) lipid signaling does not modulate 'proliferation' but instead modulates sensitivity to chemotherapy. The authors will need to perform experiments at minimum on their FASTS vs. 'other' cell lines to distinguish between these two. Knockdown of key genes such as CHP1 and GPAT4 should also be performed to establish whether the change in hazard ratio associated with these genes is related to one or the other (or both) mechanisms.

2) Along these lines, further characterisation of FASTS vs. non-FASTS lines is needed, related to changes in basal proliferation activity, and association/dependence of these features on FASTS gene expression/knockdown in these lines. Does expression of FASTS genes change with age of the lines or in patient cohorts?

Response to Reviewers' Comments

We are pleased that the reviewers found our work to be generally of interest. We would note at the outset that we have substantially revised this text to update the method that we used to classify Proliferation Suppressor Genes (PSG). Briefly, we have moved from a randomization approach to a Gaussian mixture modeling approach to generate the background distribution from which we then calculate gene-level Z scores. The new process is described in the text and figures around Figure 1. The result is still a Z-score, where positive Z-scores indicate PSG.

Although this method is overall more accurate and robust, it does not change the flow or the results of the paper; only the content of individual figure panels is updated to reflect the new “mixed Z score” approach. One material change is the addition of the PCGF1 chromatin remodeling gene to the fatty acid synthesis cluster detected in AML cells. The overall tightening of this cluster carries into the tumor genomics analysis, where four of our genes show negative hazard ratios in the TARGET cohort and a cluster of these patients with high gene expression shows markedly improved overall survival.

Despite this strong survival signature, we cannot infer that the lipid sensitivity phenotype is present in vivo. We have no tumor or primary cell data – both of which would be beyond the scope of this paper (which is already quite broad). We instead clarify the key message of the paper as follows: the novel gene signature is discovered in our analysis; its biochemical and genetic predictions are validated in vitro and in silico; and these genes carry a strong signature in tumor genomics data, potentially indicating a novel subtype of AML. The characterization of this putative subtype – whether it involves the same lipid metabolism shift -- is left for further study.

Specific reviewer responses are noted below.

Reviewer #1 (Remarks to the Author): Expert in bioinformatics and CRISPR screening

This study seeks to develop a systematic approach to classify the suppressors of proliferation genes whose knockout provides a proliferation advantage in vitro. Whole genome CRISPR knockout genetic screens have had a profound effect on the cancer functional genomics and recently large functional genomic screens in cancer cell lines become available especially in the form of Cancer Dependency Map (DepMap). Although essential gene phenotypes are examined in extensive detail there is not really much research on the proliferation suppressor phenotype. Here in this paper the authors aim to develop a framework to examine this phenotype in more detail. Using this analysis framework a network of 103 genes in 22 modules are identified. One of these modules contain several genes from glycerolipid biosynthesis pathway and operates exclusively in a subset of AML lines. A Cas12a-mediated genetic interaction screen is used to confirm the underlying differences in the lipid biosynthesis genetic interaction network between two AML cell lines. Using this screen a novel genetic interaction is identified between GPAT4 and CHP1 which has a clinical relevance on the survival.

This is a really interesting study where proliferation suppressor genes in CRISPR knockout genetic screens are performed. They analysed all the cell lines from Broad DepMap set and identified tumour suppressor genes. From this general analysis the paper focused on a novel module that is associated with fatty acid and lipid biosynthesis pathways. From here on the paper explores a subset of AML cell

lines and identified a genetic interaction that has a clinical relevance. I am quite supportive of the goals of this study in principle where you transform the results of a general analysis of CRISPR knockout genetic screens to clinically relevant conclusion. However the paper is quite difficult to follow where the objective of the study is changing constantly and jumping from one result to another without a conclusion. There are also some issues about the results and the conclusions of the paper.

The paper is overall is written quite well and the methods are explained in detail, but there are some typos and grammatical errors in the paper including the supplementary part. Also there are some issues with the Figure names in some places.

Thank you for the positive comments. We have attempted to address the grammar and typographical errors in this revision.

The proliferation suppressor genes (PSG) are identified using a data driven approach where the mean LFC of gRNAs targeting a gene are compared to a variance normalized null distribution. The null distribution is achieved by shuffling the labels of the guide level LFCs to calculate gene level mean fold change. This is repeated 1000 times. Although the method is not a novel approach, it is convincing. The results are compared to a set of known tumour suppressors. The change in the recall criteria is not justified convincingly. The study focuses on the tumour suppressors that identified in the analysis but there is no mention of the novel tumour suppressors that are identified in the analysis. In fact study focuses on 58 tumour suppressors that overlap with the COSMIC TSGs but there is no information about the other TSGs detected by this method. This study is applying this method on 563 cell lines and only focuses on known TSGs. All the mutation and gene expression analysis is performed on the 58 known TSGs. I think the analysis should be extended to all results of the shuffling method. Although gene expression and silent mutations are inspected there is no information on the copy number of these genes which might be interesting too.

Comparative analyses on the 58 known TSG were performed as validation steps to demonstrate that the proliferation suppressor genes (PSG) we identify in the CRISPR data *in vitro* are in fact very often tumor suppressor genes *in vivo*. We have updated supplemental figure 2 to include CN comparisons of PS vs non PS calls of COSMIC TSGs, and for all other non-COSMIC TSGs proliferation suppressor observations. As for going beyond COSMIC TSG, we specifically mention PDCD10 in the text, and discuss in detail the FAS genes as proliferation suppressor observations that our methodology picks up. None of these genes are considered tumor suppressors in the cancer gene census by COSMIC. In fact the entire second half of the paper is about characterizing a novel module of PSG and their putative role as TSG.

Another issue with the study is the detailed analysis of the CRISPR screen dataset from Sanger. There is a detailed explanation of why the screen is not used. It seems that part is a little bit more detailed and may be shortened as the data is not used at the end. In the supplementary there is a really nice explanation about why it is not used. However in the supplementary it seems that it might be possible to adjust for the time difference and maybe the data might be integrated. It would be also interesting to check the PSGs that are identified by the shuffling method in this dataset and check if they cannot be detected at all.

The nature of this discrepancy is rooted in the difference in fitness effects between essential genes (typically strong) and proliferation suppressors (typically weak). Weak fitness phenotypes require more doublings to be differentiated from the distribution of null-

phenotype knockouts. In the supplementary information and in Supplementary Figure 3 we show that PSG in the 21-day Avana screens show a strong bias toward positive Z scores in the 14-day Score screens, validating the increased fitness from these gene knockouts. However the Z-scores of these genes are typically not significant in the Score data (e.g. in Supp Fig 2A, the Avana PSG have mean Z-scores $\sim+2$ in the Score data), preventing us from using this data set as a discovery source for all but a few of the strongest hits.

The correlation network for the PSG is quite interesting and novel idea. However limiting the networks to only PSG might be a little limiting. It would be interesting to include essential genes to the networks to see how it affects the connectivity. Another limitation is to use only the PSGs that are observed in two cell lines. This will make it hard to investigate rare PSGs. Overall co-occurrence PSG networks is an interesting and novel idea. It would be also interesting to compare these networks with the PPI networks or their shortest path distance over the PPI networks.

We agree that further network exploration would be interesting, and our work characterizing the network rewiring of FASTS cells and identifying positive and negative correlations that are amplified in AML cells (Figure 3) – which includes both essential genes and PSG -- shows the potential of this approach. We look forward to community engagement with this dataset to pursue novel findings beyond what we are able to describe in one manuscript. With regard to other networks, in Supplementary Figure 4 we compare our PSG network with the broader HumanNet functional interaction network, with these approaches confirming that co-occurrence in the PSG network is analogous to correlated knockout fitness for essential genes and implies co-functionality.

Although some information is given on these modules there is not a good characterization of the modules other than the fatty acid pathways. Is this due to the fact that they are not interesting or they are well known pathways or there are not enough genes in the modules. It would be interesting to look at the pathway enrichment of all the modules together.

In our interpretation, each module represents its own unique process for constraining the growth rate of a particular class of cells. We look forward to deeper dives into the mechanisms and context specificity of these processes, but we already have quite a long paper focusing on one such module.

The validation of the genetic interaction identified in the subset of AML cell lines using the Cas12a-mediated genetic interaction is quite exciting. However the method that are used for detecting the gene interaction scores is not very well described. In the study only two cell lines are used and it is not clear if there are any replicates. Also they have a 14 day and 21 day samples but the 14 day sample results are not mentioned in the results. It would confirm the difference in the previous section related to the time course of CRISPR knockout genetic screens. Also it would be more effective to run the screen on all the 4 AML cell lines that are FASTS and 4 non FASTS cell lines.

We have attempted to clarify the genetic interaction scoring scheme that we used, but we note that it is very similar to the one used in Horlbeck et al (Horlbeck, Cell 2018). We have included the 14-day results in Supplementary Figure 6 but note that, as with the comparison between the Avana and Score data, the 14-day timepoint is less informative than the 21-day.

The clinical relevance part is quite convincing. It will require a little bit more inspection on the mutational signature that drive the phenotype. The gene expression observation is quite important clinically and important. It would be worth checking the same expression profile in other cancer types and tissues in order to understand why it is specific to a subset of AML.

We thank the reviewer for this comment, and acknowledge that while the genetic signature that we discover with these *in vitro* analyses points us to a clinically relevant gene expression signature *in vivo* with a very strong prognostic signature, the hypothesis that the lipid sensitivity we observe *in vitro* also exists *in vivo* is not yet validated. We have attempted to clarify this message in the text.

There some small issues with the paper and the results section can be extended a little bit more. However it is quite novel paper in terms of the development of the study. The differential correlation of shuffled Z-scores can be applied for other cell line sub-groups. The paper might benefit from a general figure that shows the whole process from the initial DepMap analysis to the clinical analysis. Overall the paper is written well except some small issues which can be addressed by going over the text and the methods are described in detail. All the software used in the study as well as their versions are identified clearly. Although the github pages are not accessible it seems the scripts are there. The experimental protocols are described in detail in the supplementary and tables are included in the submission.

We thank the reviewer for these comments, and will open the github repository to the public immediately upon acceptance. Additionally, final data sets will be uploaded to Figshare pre-publication and links will be included in the final manuscript. We are firmly committed to open, reproducible research practices.

Reviewer #2 (Remarks to the Author): Expert in AML and metabolism

The manuscript by Lenoir and colleagues describes a new method for analyzing CRISPR screen data to identify genes that suppress cancer cell line growth. Utilizing this method, the authors identify that fatty acid synthesis acts as a tumor suppressor in a subset of AML cell lines. In a pediatric patient cohort elevated expression of genes involved in fatty acid metabolism were shown to be associated with a survival benefit. While the results are intriguing several approaches could be taken to strengthen the overall conclusions. Further, additional explanation of how this method could be used in future screens would be helpful (see comment 1).

1. The CRISPR screen analysis was not reproducible between the two datasets analyzed (Behan et al. and Avana) using the authors new analysis method. Dempster et al. Nature Communications, 2019 reported that these datasets had large agreement between them despite differences in the experimental design. The authors contribute the difference in their method to a shorter assay time. Do the authors think that their method is limited in use to longer-term screening approaches?

As we note above in response to Reviewer #1, the nature of this discrepancy is rooted in the difference in fitness effects between essential genes (typically strong) and proliferation suppressors (typically weak). Weak fitness phenotypes require more doublings to be differentiated from the distribution of null-phenotype knockouts. As with many similar analyses, there is a tradeoff between effect size and statistical power. In short: yes, for smaller effect sizes, longer screening assays are required.

2. The authors caution the use of FASN inhibitors. Can the authors show FASN inhibitors increases cell proliferation and viability in the FASTS and not other AML cell lines?

We acknowledge that observing an *in vitro* phenotype does not necessarily imply that the same phenotype (in this case, lipid sensitivity or susceptibility to FASN inhibitors) exists *in vivo* or in patients. However, we are quite confident that the FASTS module which we discover from *in vitro* genetics corresponds to the FASTS prognostic signature in the TARGET cohort. Confirming the mechanistic basis of this tumor suppressor phenotype *in vivo* or in patients, or recruiting collaborators to join this effort, has been beyond our capabilities during these months of COVID restrictions. We have softened the language around whether the FASTS lipid-sensitive phenotype is present in tumors and instead focused on the presence of the prognostic value of the expression signature of the FASTS genes, leaving the mechanistic exploration of this signature for future studies.

3. The authors should show if the FASTS signature correlates with sensitivity to fatty acids in primary AML specimens.

Unfortunately, we do not have a mechanism yet for identifying FASTS cells from primary AML specimens, which would be required for doing this experiment. This is clearly a question that we would like to answer in follow-up studies of the FASTS phenotype.

4. Does the FASTS signature correlate with age in the different AML patient cohorts? Can the authors provide an explanation for differences observed in the adult vs pediatric patient cohorts?

The FASTS *in vitro* genetic signature is not associated with differential expression of FASTS cluster genes in cell lines and gene expression of FASTS cluster genes is not associated with age in any of the public tumor data. We have included these comparisons in Supplementary Figure 8. We have a number of hypotheses about the emergence of this phenotype, especially since the re-analysis and discovery of an epigenetic regulator as part of this cluster, but testing these hypotheses will require follow-on research projects that build on the initial observations described here. We stand by the conclusion that a novel approach to CRISPR screen analysis reveals a set of genes associated with lipid metabolism that confers a growth advantage to a subset of AML cells, and that overexpression of these genes is associated with survival advantage in AML patients. Further mechanistic studies are absolutely warranted but are, in our view, beyond the scope of a paper which already integrates a number of disparate approaches to this topic.

Reviewer #3 (Remarks to the Author): Expert in AML and metabolism

This is a very interesting computational study based on identifying gene modules associated with fitness changes in CRISPR screens of AML cells. The authors have defined a very interesting pipeline for analyzing these data and making potential predictions about AML outcome. The biological validation of the computational approach is in its current form however not sufficient to rule out alternative hypotheses and support the authors' conclusions and the translational potential of the work.

In figure 6, the survival probability shown in figures E and F could be explained by at least two alternative hypotheses - 1) the leukemia is less 'proliferative' due to lipid signaling or 2) lipid signaling does not modulate 'proliferation' but instead modulates sensitivity to chemotherapy. The authors will need to perform experiments at minimum on their FASTS vs. 'other' cell lines to distinguish between these two. Knockdown of key genes such as CHP1 and GPAT4 should also be performed to establish whether the change in hazard ratio associated with these genes is related to one or the other (or both) mechanisms.

We agree that no mechanistic explanation behind increased patient survival is supported by any data that we present. To be more precise, we have softened the language around our conclusions in the manuscript, as described in detail in our responses to Reviewer 1 and Reviewer 2: “We stand by the conclusion that a novel approach to CRISPR screen analysis reveals a set of genes associated with lipid metabolism that confers a growth advantage to a subset of AML cells, and that overexpression of these genes is associated with survival advantage in AML patients. Further mechanistic studies are absolutely warranted but are, in our view, beyond the scope of a paper which already integrates a number of disparate approaches to this topic.”

2) Along these lines, further characterisation of FASTS vs. non-FASTS lines is needed, related to changes in basal proliferation activity, and association/dependence of these features on FASTS gene expression/knockdown in these lines. Does expression of FASTS genes change with age of the lines or in patient cohorts?

In brief: there is no expression signal in the in vitro data, and there is no age-related change in FASTS cluster genes in the tumor data. As we note in a response to Reviewer 2: “The FASTS *in vitro* genetic signature is not associated with differential expression of FASTS cluster genes in cell lines and gene expression of FASTS cluster genes is not associated with age in any of the public tumor data. We have included these comparisons in Supplementary Figure 8.”

REVIEWERS' COMMENTS

Reviewer #1 (Remarks to the Author):

Authors have substantially revised the paper and the results are pretty much consistent with the previous version of the paper. This is a really interesting study where proliferation suppressor genes in CRISPR knockout genetic screens are performed. From their general analysis the paper focused on a novel module that is associated with fatty acid and lipid biosynthesis pathways and identified a genetic interaction that has a clinical relevance. This is quite an important results and proof of principle that general analysis of CRISPR knockout genetic screens can lead to important clinically relevant conclusions.

The authors included the copy number comparison of PS and non-PS calls from COSMIC TSGs in the supplementary table and addressed the issue related to non-COSMIC TSGs. They also explained the scope of the paper more clearly. The lipid sensitivity that is observed in the vitro is not yet validated in vivo and this message is cleared in the paper. Also I agree that exploring all the modules is beyond the scope of this paper and the results of this paper will be an important contribution to the exploration of these modules and development of new system biology approaches.

The authors also screened one AML cell line from the FASTS subset, and a second one with no FAS phenotype, collecting samples at 14 and 21 days after transduction and addressed one of my main issues.

Overall the revised version of the paper addressed most of the issues related to the paper. Also some of the suggestions were beyond the scope of this paper and can be explored further in the follow-up papers especially related to the other modules and the exploration of other networks and other system biology models. The language of the paper is also changed and cleared with respect to the explanation of the mechanism behind this module. It is more clear that there is no evidence in vivo and more exploration of this hypothesis is needed. It is also more clear the limitations of the paper related to the explanation and validation of the mechanism. However this is an excellent demonstration of the use of computation methods to understand clinically important signals. I think this method shows the power of the CRIPSR screens to transfer this knowledge to clinic.

Reviewer #2 (Remarks to the Author):

The authors have made significant revisions to the text and description of the methodology which has improved the paper. Functional validation of the findings in patient specimens would greatly impact the overall significance of this work. However, this reviewer acknowledges that it is reasonable for these experiments to be considered outside of the scope of the current manuscript.

Reviewer #3 (Remarks to the Author):

We feel the authors have sufficiently addressed our concerns, given the enhancements to the manuscript and toning down claims related to the biology and therapeutic implications.

Response to Reviewers' Comments

Reviewer #1 (Remarks to the Author):

Authors have substantially revised the paper and the results are pretty much consistent with the previous version of the paper. This is a really interesting study where proliferation suppressor genes in CRISPR knockout genetic screens are performed. From their general analysis the paper focused on a novel module that is associated with fatty acid and lipid biosynthesis pathways and identified a genetic interaction that has a clinical relevance. This is quite an important results and proof of principle that general analysis of CRISPR knockout genetic screens can lead to important clinically relevant conclusions.

The authors included the copy number comparison of PS and non-PS calls from COSMIC TSGs in the supplementary table and addressed the issue related to non-COSMIC TSGs. They also explained the scope of the paper more clearly. The lipid sensitivity that is observed in the vitro is not yet validated in vivo and this message is cleared in the paper. Also I agree that exploring all the modules is beyond the scope of this paper and the results of this paper will be an important contribution to the exploration of these modules and development of new system biology approaches. The authors also screened one AML cell line from the FASTS subset, and a second one with no FAS phenotype, collecting samples at 14 and 21 days after transduction and addressed one of my main issues.

Overall the revised version of the paper addressed most of the issues related to the paper. Also some of the suggestions were beyond the scope of this paper and can be explored further in the follow-up papers especially related to the other modules and the exploration of other networks and other system biology models. The language of the paper is also changed and cleared with respect to the explanation of the mechanism behind this module. It is more clear that there is no evidence in vivo and more exploration of this hypothesis is needed. It is also more clear the limitations of the paper related to the explanation and validation of the mechanism. However this is an excellent demonstration of the use of computation methods to understand clinically important signals. I think this method shows the power of the CRIPSR screens to transfer this knowledge to clinic.

We thank the reviewer for these comments

Reviewer #2 (Remarks to the Author):

The authors have made significant revisions to the text and description of the methodology which has improved the paper. Functional validation of the findings in patient specimens would greatly impact the overall significance of this work. However, this reviewer acknowledges that it is reasonable for these experiments to be considered outside of the scope of the current manuscript.

We thank the reviewer for these comments

Reviewer #3 (Remarks to the Author):

We feel the authors have sufficiently addressed our concerns, given the enhancements to the manuscript and toning down claims related to the biology and therapeutic implications.

We thank the reviewer for these comments